# Uncovering Latent Communication Patterns
# in Brain Networks via Adaptive Flow Routing

**Tianhao Huang** [1]   **Guanghui Min** [1]   **Zhenyu Lei** [1]   **Aiying Zhang** [1]   **Chen Chen** [1]

## Abstract

Unraveling how macroscopic cognitive phenotypes emerge from microscopic neuronal connectivity remains one of the core pursuits of neuroscience. To this end, researchers typically leverage multi-modal information from structural connectivity (SC) and functional connectivity (FC) to complete downstream tasks. Recent methodologies explore the intricate coupling mechanisms between SC and FC, attempting to fuse their representations at the regional level. However, while these approaches do incorporate useful neuroscientific observations, they predominantly operate at a topological or architectural level and lack a principled formulation grounded in neural communication dynamics. Consequently, they are limited in quantifying how information is actually routed between neural regions, and thus cannot fully explain why SC and FC exhibit dynamic states of both coupling and heterogeneity. In this paper, we formulate multi-modal fusion through the lens of neural communication dynamics and propose the **A**daptive **F**low **R**outing **Net**work (AFR-Net), a physics-informed framework that models how structural constraints give rise to functional communication patterns, enabling interpretable discovery of critical neural pathways. Extensive experiments demonstrate that AFR-Net significantly outperforms state-of-the-art baselines. The code is available at https://github.com/Skyyyy0920/AFR-Net.

## 1. Introduction

The human brain is intricately organized as a complex network, where cognitive functions emerge from the dynamic interactions among billions of neurons (Avena-Koenigsberger et al., 2018). In the realm of connectomics, this organization is typically characterized by two complementary modalities: structural connectivity (SC), which maps the physical anatomical tracts (e.g., white matter streamlines) serving as the hard-wired infrastructure, and functional connectivity (FC), which captures the statistical temporal correlations of neuronal activities representing information traffic (Friston, 1994; Friston et al., 1993; Sporns et al., 2005; Le Bihan et al., 2001; Pang et al., 2023). Deciphering the complex coupling mechanism between these two modalities—specifically, how the static SC scaffold constrains and shapes the dynamic FC patterns (Honey et al., 2009; Goñi et al., 2014; Sarwar et al., 2021; Neudorf et al., 2022; Ran et al., 2024)—remains an important problem in computational neuroscience and is critical for identifying biomarkers of neurological disorders (Park & Friston, 2013; Li et al., 2021; Segal et al., 2023; Hu et al., 2024).

Current research efforts seek to elucidate the complex coupling mechanisms between SC and FC, and have made substantial contributions toward more effective fusion and integrative modeling of these two modalities. Early studies primarily focused on representing FC and SC using GNN-based approaches (Kipf & Welling, 2017), and subsequently fusing the two modalities at the regional level through relatively simple fusion strategies, such as concatenation (Sebenius et al., 2021; Wang et al., 2025; Li et al., 2021), weighted summation (Zhang et al., 2021), or self-attention mechanisms (Zhang & Shi, 2020; Zuo et al., 2023; Oota et al., 2024). However, despite their strong performance in feature extraction, these data-driven methods tend to over-rely on static topological structures and mainly capture local homophily. This theoretical limitation hinders these models from adequately explaining the pronounced discrepancies between structural connections and synchronized functional activity, commonly referred to as the regional heterogeneity problem. Meanwhile, a substantial body of literature indicates that the relationship between FC and SC is far from one-to-one mapping and is instead highly complex (Koch et al., 2002; Lim et al., 2019; Suárez et al., 2020). At the regional level, FC and SC may be tightly coupled in certain brain regions, while exhibiting weak coupling in others. To address these challenges, re-

[1]Department of Computer Science, University of Virginia, Charlottesville, USA. Correspondence to: Chen Chen <zrh6du@virginia.edu>.

*Proceedings of the $43^{rd}$ International Conference on Machine Learning*, Seoul, South Korea. PMLR 306, 2026. Copyright 2026 by the author(s).

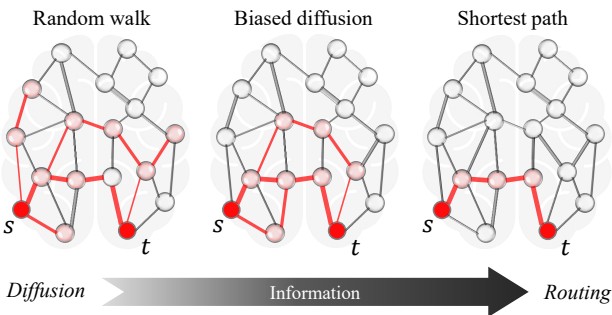

Random walk  Biased diffusion  Shortest path

*Diffusion*  Information  *Routing*

*Figure 1.* Neural communication models can be categorized along a spectrum ranging from diffusion to routing, based on the extent of global topological information available to the network. We hypothesize that biological neural communication operates in an intermediate regime, utilizing multiple alternative paths to ensure robustness and efficiency. In the illustration, the thickness of each path represents its information flow capacity, which is derived from the reciprocal of the connection strength.

searches such as RH-BrainFS (Ye et al., 2023) address the regional heterogeneity of SC-FC coupling by introducing fusion bottlenecks. Concurrently, increasing evidence from network control theory suggests that neural signals propagate via multiple parallel pathways to ensure robustness and efficiency, a process better modeled by diffusion dynamics or circuit theory rather than simple topological hops (Griffa et al., 2023; Neudorf et al., 2023). Recent works like NeuroPath (Wei et al., 2024b) introduce the concept of "topological detours" to model indirect structural support for functional links, while EDT-PA (Sheng et al., 2025) models high-order structural dependencies via adaptive diffusion and aligns structural and functional connectomes through pattern-specific optimal transport.

While acknowledging that neural signaling traverses multiple pathways beyond direct connections or shortest paths, these approaches predominantly tackle the problem from a *topological or architectural* perspective (e.g., detour masks in NeuroPath, fusion bottlenecks in RH-BrainFS). We emphasize that this is a matter of *level of abstraction* rather than a lack of neuroscientific grounding: circuit-theoretic and diffusion-based flow models are themselves the dominant neuroscience framework for characterizing brain *communication dynamics* (Avena-Koenigsberger et al., 2018; Seguin et al., 2020; 2023). The distinction we draw is between topological methods, which model *which* structural pathways support a functional link, and communication-dynamics methods, which quantify *how much* information is routed along each pathway. Because prior methods do not explicitly model communication dynamics, they are limited in quantifying the magnitude of information transmission across distinct routes and therefore cannot readily distinguish the critical signaling pathways that carry the majority of the information flow. In reality, we posit that authen-tic neural signaling operates within an intermediate regime between stochastic random walks and deterministic shortest paths (Bullmore & Sporns, 2009; Avena-Koenigsberger et al., 2018; 2019; Amico et al., 2021; Seguin et al., 2018; 2020; 2023), as illustrated in Figure 1. On this basis, we propose **AFR-Net** (**A**daptive **F**low **R**outing **Net**work), a novel physics-informed framework that explicitly models brain network coupling as a global information flow process. Drawing inspiration from transport theory and electric circuit analogs, we conceptualize the structural connectome as a dynamic flow network with adaptive capacities, rather than a static graph. Unlike previous black-box fusion strategies, AFR-Net introduces a differentiable flow routing module that mathematically deduces how information "flows" through the structural cables to generate the observed functional correlations. By solving a global equilibrium governed by a regularized graph Laplacian, our model identifies the critical transmission pathways—highlighting edges with high "information load"—and uses these latent patterns to guide the message-passing process.

Specifically, AFR-Net incorporates three key innovations aligned with the nature of neural communication. First, we introduce a *Physics-Informed Graph Construction* phase, which draws inspiration from electrical circuit theory, specifically the computation of effective resistance, to construct a dynamic flow network with learnable edge capacities based on the topological structure. Second, we derive a *Differentiable Information Flow Solver*. By modeling the brain as a resistor network, we compute a closed-form solution for edge-level information traffic, explicitly characterizing how global functional demands drive flow through the structural constraints. Finally, a *Pattern-Guided Aggregation* layer leverages these mathematically derived flow intensities to bias the model's attention toward biologically meaningful pathways. We validate AFR-Net on multiple datasets, including Adolescent Brain Cognitive Development Study and Parkinson's Progression Markers Initiative. The extensive experiments demonstrate that our method not only achieves state-of-the-art performance but, more importantly, offers superior interpretability by uncovering the specific neural circuits that underpin pathological deviations.

Our main contributions are summarized as follows:

- We propose a mechanism-driven framework that formulates multimodal brain network fusion as a latent flow routing problem, offering a mathematically grounded explanation for SC-FC coupling.

- We introduce AFR-Net, which integrates physics-informed graph construction with a differentiable closed-form flow solver to capture the global, multi-path nature of neural communication.

- AFR-Net significantly outperforms existing baselines

across two large-scale benchmarks spanning four binary disease-classification tasks. Furthermore, it identifies disease-specific routing patterns that align with clinical neuroscientific findings, demonstrating its potential as a tool for biomarker discovery.

## 2. Related Work

**Brain Network Analysis**   The application of deep learning to connectomics has evolved from applying generic graph algorithms to designing brain-inspired architectures. Early works adapted standard GNNs, such as GCN (Kipf & Welling, 2017) and GAT (Velickovic et al., 2018), to classify functional or structural connectomes. While effective, these models treat brain regions as interchangeable nodes, ignoring the unique topological properties of neural circuits. To address this, specialized architectures emerged: BrainGNN (Li et al., 2021) introduced ROI-aware pooling to preserve anatomical identity, while Transformer-based approaches like Brain Network Transformer (Kan et al., 2022) and AL-TER (Yu et al., 2024) leverages self-attention to capture long-range dependencies essential for cognitive integration. Recognizing that structural and functional views are complementary, recent research has shifted towards multimodal fusion. The core challenge lies in modeling the complex coupling between the static structural connectivity scaffold and dynamic functional connectivity signals (Park & Friston, 2013). Cross-GNN (Yang et al., 2024) proposes a mutual learning framework to regularize latent representations across modalities, while Triplet (Zhu et al., 2022) employs multi-scale attention to extract hierarchical features. More recently, Ye et al. (2023) propose RH-BrainFS to tackle regional heterogeneity by introducing fusion bottlenecks to model indirect interactions. Bian et al. (2023) propose an adversarially trained persistent homology-based GCN that integrates global topological invariants from persistent homology with local graph features, utilizing a clinical prior-guided adversarial training strategy to enhance robustness in brain disease diagnosis. These methods make meaningful progress on regional heterogeneity, but predominantly at the *topological* level (e.g., RH-BrainFS via fusion bottlenecks, NeuroPath via detour masks). What remains comparatively underexplored is a complementary treatment at the level of *communication dynamics*: explicitly quantifying how much information each structural pathway carries to support the observed functional dependencies. Without such a process-level formulation, these methods learn to map inputs to labels via statistical correlations rather than the underlying biological routing process, which limits their ability to explain the mechanistic "why" behind their predictions.

**Communication Dynamics and Flow Routing**   Network neuroscience posits that cognitive functions emerge from dynamic communication processes unfolding on the structural connectome (Park & Friston, 2013; Avena-Koenigsberger et al., 2018; Fakhar et al., 2025). Two dominant paradigms exist: *routing*, where signals travel via selective shortest paths, and *diffusion*, where information spreads stochastically or flows based on global network conductance (Graham, 2014; Mišić et al., 2015; Seguin et al., 2020; 2023). While shortest path efficiency is a standard metric, growing evidence suggests that brain communication utilizes parallel, redundant pathways to ensure robustness (Sporns et al., 2005; Avena-Koenigsberger et al., 2018). NeuroPath (Wei et al., 2024b) recently attempted to model "topological detour" but remains limited to heuristic masking. Wei et al. (2024a) proposes detour connectivity to characterize the SC-FC coupling by enumerating indirect structural pathways that sustain functional interactions. Most existing deep learning models essentially overfit to the routing efficiency hypothesis (via message passing on direct edges), neglecting the broader "flow-based" nature of neural signaling. Our proposed method AFR-Net distinguishes itself by mathematically embedding this flow perspective into a deep learning framework. Unlike prior works, we do not simply fuse SC and FC features; instead, we treat SC as a flow network with learnable capacities and FC as the demand driving the flow, solving for a global equilibrium state that aligns with physical diffusion laws.

## 3. Preliminaries

### 3.1. Problem Formulation

We define the Multimodal Brain Network Analysis task over a dataset $\mathcal{D} = \{\mathcal{S}_1, \mathcal{S}_2, \ldots, \mathcal{S}_K\}$, where $K$ denotes the total number of subjects. Each sample $\mathcal{S}_k = (\mathcal{G}_{sc}^{(k)}, \mathcal{G}_{fc}^{(k)}, y^{(k)})$ consists of a structural connectivity graph, a functional connectivity graph, and a corresponding label $y^{(k)} \in \{0, 1, \ldots, C - 1\}$, where $C$ represents the number of diagnostic classes (e.g., $C = 2$ for binary diagnosis).

Formally, we represent the brain network as a weighted undirected graph $\mathcal{G} = (\mathcal{V}, \mathbf{A}, \mathbf{X})$, where $\mathcal{V} = \{v_1, \ldots, v_N\}$ is the set of $N$ nodes corresponding to specific brain regions, and $\mathbf{X} \in \mathbb{R}^{N \times d_x}$ is the node feature matrix encoding regional attributes. The connectivity structure is distinguished by the symmetric adjacency matrix $\mathbf{A}$:

- Structural connectivity graph $\mathcal{G}_{sc} = (\mathcal{V}, \mathbf{A}_{sc}, \mathbf{X})$, the adjacency matrix $\mathbf{A}_{sc} \in \mathbb{R}_{\geq 0}^{N \times N}$ represents the physical anatomical connections (e.g., white matter fiber streamlines) between regions.

- Functional connectivity graph $\mathcal{G}_{fc} = (\mathcal{V}, \mathbf{A}_{fc}, \mathbf{X})$, where the adjacency matrix $\mathbf{A}_{fc} \in \mathbb{R}^{N \times N}$ captures the statistical dependencies (e.g., temporal correlations) observed among dynamic regional neural activities.

Given this multimodal input, our objective is to learn a

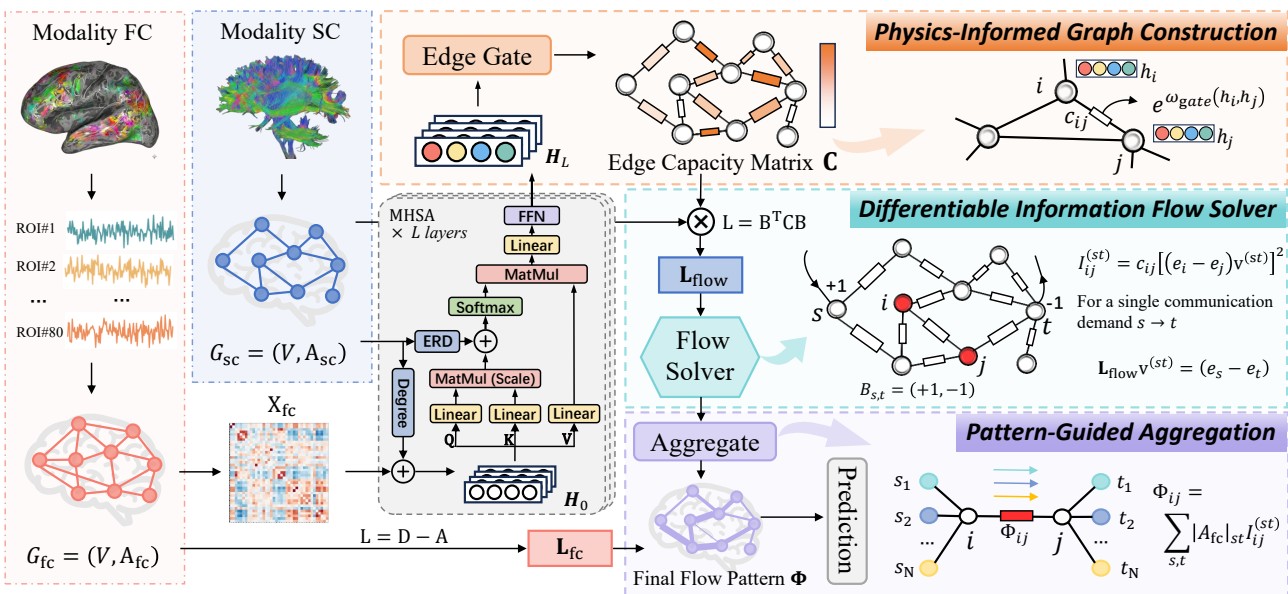

*Figure 2.* The overall framework of our proposed method AFR-Net. The model explicitly fuses two modalities of structural and functional connectivity via three integrated phases: (1) Physics-Informed Graph Construction, which initializes structure-aware node representations using effective resistance distance (ERD) and constructs a dynamic flow network with learnable edge capacities calculated by edge gate; (2) Differentiable Information Flow Solver, which computes a closed-form solution for global information traffic driven by functional demands ($\mathbf{L}_{fc}$) through structural constraints ($\mathbf{L}_{flow}$); and (3) Pattern-Guided Aggregation, which utilizes the learned flow patterns to guide message passing for downstream classification tasks.

mapping function $f : (\mathcal{G}_{sc}, \mathcal{G}_{fc}) \to y$ that accurately predicts the subject's downstream task label.

### 3.2. Graph Laplacian

The graph Laplacian is traditionally defined as $\mathbf{L} = \mathbf{D} - \mathbf{A}$, where $\mathbf{D}$ is the degree matrix. To model information flow as a physical process, we utilize the algebraic representations of graphs. We define the node-edge incidence matrix $\mathbf{B} \in \{-1, +1, 0\}^{M \times N}$, where $M$ is the count of unique undirected edges in the graph. For each edge $e_m = (i, j) \in \mathbf{A}$, we assign an arbitrary orientation (e.g., $i \to j$) yielding entries $B_{mi} = 1$ and $B_{mj} = -1$. This allows us to express the potential difference across an edge using the standard basis vectors. Let $\mathbf{e}_i \in \mathbb{R}^N$ be the one-hot indicator vector for node $v_i$. The vector $(\mathbf{e}_i - \mathbf{e}_j)$ corresponds to the $m$-th row of $\mathbf{B}$, mapping node potentials to edge gradients. Crucially, analogous to reference directions in circuit theory, this orientation is merely a mathematical convention required to define discrete gradient operators. Since our flow intensity metric is derived from the squared potential difference (akin to power dissipation $P = I^2 R$), it captures the magnitude of information load. Consequently, the metric remains invariant to the initial edge orientation, ensuring the final representation reflects the connection's usage intensity rather than a vector direction.

Let $\mathbf{C} = \text{diag}(c_1, \dots, c_M)$ denote the diagonal matrix encapsulating the learned weight of edges. The weighted graph Laplacian $\mathbf{L_C}$ can then be rigorously decomposed as:

$$\mathbf{L_C} = \mathbf{B}^\top \mathbf{C} \mathbf{B} = \sum_{(i,j) \in \mathbf{A}} c_{ij} (\mathbf{e}_i - \mathbf{e}_j)(\mathbf{e}_i - \mathbf{e}_j)^\top. \quad (1)$$

This decomposition is central to our framework, as it mathematically links the local edge capacity ($c_{ij}$) to the global diffusion dynamics on the neural substrate.

### 4. Method

In this section, we elaborate on the proposed method AFR-Net, a physics-informed framework designed to fuse structural and functional connectivity (Figure 2). The methodological pipeline is organized into three sequential stages: First, we establish the biologically plausible environment by constructing a parameterized flow network and initializing structure-aware node features (Section 4.1). Next, we derive the theoretical core of our framework—a differentiable closed-form solver that computes global information flow equilibrium driven by functional demands (Section 4.2). Finally, we present the pattern-guided mechanism that leverages these learned routing patterns to aggregate features for disease diagnosis (Section 4.3).

**Intuition.** Before the formal derivations, we give the high-level picture. AFR-Net treats the structural connectome as an electrical circuit: anatomical tracts are wires with (learnable) conductances, and every pair of regions that

needs to communicate—as indicated by their functional connectivity—injects a unit of "current" into the network. Solving this circuit yields, for each anatomical edge, the total amount of traffic it must carry to satisfy all functional demands; this edge-level traffic map is exactly the latent communication pattern we seek. Heavily trafficked edges are then used to bias message passing for the downstream task. Importantly, the circuit equations are *not* free parameters: they impose a global physical constraint, and only a small set of components (Table 4, Appendix A) is actually learned, with everything else fixed by the data.

### 4.1. Physics-Informed Graph Construction

To establish a biologically plausible foundation for modeling neural communication, we first construct a physics-informed graph environment. This process comprises two key steps: explicitly encoding the geometric positioning of brain regions (nodes) and dynamically determining the transmission capacity of anatomical connections (edges).

#### 4.1.1. STRUCTURE-AWARE NODE INITIALIZATION

To simulate neural communication, node representations must explicitly encode their geometric position within the structural scaffold. Standard GNNs often utilize local message passing that ignores the global topology of the brain. To address this, inspired by graphormer (Ying et al., 2021), we propose a structure-aware flow encoder that replaces topological heuristics with effective resistance distance (ERD). Unlike Graphormer which relies on shortest path distance (SPD) to encode structural locality, in our framework, the proximity between two regions is not defined by "hops", but by their transport connectivity (the effective resistance distance)—a measure derived from electrical network analysis for graph (Ghosh et al., 2008; Black et al., 2023).

We initialize node representations by augmenting projected features with a structural prior:

$$\mathbf{h}_i^{(0)} = \text{Linear}(\mathbf{x}_i) + \phi_{\text{deg}}(\deg(v_i)), \tag{2}$$

where $\phi_{\text{deg}} : \mathbb{N} \to \mathbb{R}^d$ is a learnable mapping function that encodes the discrete node degree into the hidden dimension.

We then replace the standard position-encoding self-attention with a resistance-biased attention. For any two nodes $v_i$ and $v_j$, the attention score $\alpha_{ij}^{(l)}$ is computed as:

$$\alpha_{ij}^{(l)} = \frac{(\mathbf{Q}^{(l)}\mathbf{h}_i^{(l-1)})^\top (\mathbf{K}^{(l)}\mathbf{h}_j^{(l-1)})}{\sqrt{d}} + \psi_{\text{res}}(R_{ij}), \tag{3}$$

where $\mathbf{Q}^{(l)}, \mathbf{K}^{(l)} \in \mathbb{R}^{d \times d}$ denote the learnable query and key projection matrices for the $l$-th layer, respectively. Here $\psi_{\text{res}}(\cdot)$ is a learnable encoder that maps the scalar effective resistance distance $R_{ij}$ to an attention-bias term; in practice

it is implemented as a two-layer MLP (hidden dimension 128, GELU activation).

Crucially, $R_{ij}$ is the effective resistance distance computed on the weighted structural Laplacian $\mathbf{L}_{\text{sc}}$:

$$R_{ij} = (\mathbf{e}_i - \mathbf{e}_j)^\top \mathbf{L}_{\text{sc}}^\dagger (\mathbf{e}_i - \mathbf{e}_j). \tag{4}$$

Here, $\mathbf{L}_{\text{sc}}^\dagger$ denotes the Moore-Penrose pseudoinverse of the structural Laplacian $\mathbf{L}_{\text{sc}}$, which is constructed using SC as edge conductances. This ensures that $R_{ij}$ naturally integrates edge features: a strong structural connection reduces the resistance between nodes, thereby increasing their attentional affinity. Unlike SPD, $R_{ij}$ decreases as the number of parallel paths increases, allowing the model to prioritize robust communication channels supported by dense wiring, aligning perfectly with the physical intuition of neural information flow (Griffa et al., 2023).

Based on the resistance-biased attention scores, the node representations are updated iteratively through $L$ Transformer layers. For each layer $l = 1 \ldots L$, the update rule follows the standard Transformer architecture with the injected structural bias:

$$\mathbf{Z}^{(l)} = \text{Norm}\left(\mathbf{H}^{(l-1)} + \text{Softmax}(\mathbf{A}^{(l)})\mathbf{H}^{(l-1)}\mathbf{V}^{(l)}\right), \tag{5}$$

$$\mathbf{H}^{(l)} = \text{Norm}\left(\mathbf{Z}^{(l)} + \text{FFN}(\mathbf{Z}^{(l)})\right), \tag{6}$$

where $\mathbf{A}^{(l)}$ is the attention score matrix with entries $\alpha_{ij}^{(l)}$, $\mathbf{V}^{(l)} \in \mathbb{R}^{d \times d}$ is the learnable value projection matrix, and $\text{FFN}(\cdot)$ denotes the Feed-Forward Network. The final output $\mathbf{H} = \mathbf{H}^{(L)} \in \mathbb{R}^{N \times d}$ constitutes the structure-aware node initialization, which serves as the input signal for the subsequent adaptive flow routing module.

#### 4.1.2. CONSTRUCTING THE ADAPTIVE FLOW NETWORK

We conceptualize the structural connectome as a dynamic flow network rather than a fixed graph. For each structural edge $e_{ij}$ connecting nodes $v_i$ and $v_j$, we assign a latent flow capacity $c_{ij}$. Analogous to conductance in physical systems, this capacity represents the edge's ability to transmit information flow.

Since the utilization of anatomical links varies by functional context, $c_{ij}$ is learned dynamically. We employ an edge gating network $\omega_{\text{gate}}$ that predicts capacity based on local node states: $c_{ij} = \exp\left(\omega_{\text{gate}}(\mathbf{h}_i, \mathbf{h}_j)\right)$. Concretely, $\omega_{\text{gate}}(\cdot)$ is a two-layer MLP (hidden dimension 64, SiLU activation) whose input is the edge feature $[\,|\mathbf{h}_i - \mathbf{h}_j|,\ \mathbf{h}_i \odot \mathbf{h}_j,\ (\mathbf{A}_{\text{sc}})_{ij},\ (\mathbf{A}_{\text{fc}})_{ij}\,] \in \mathbb{R}^{2d+2}$, and the exponential ensures a strictly positive capacity $c_{ij} > 0$.

The global topology of this flow network is governed by the weighted graph Laplacian. To ensure numerical stability and

strict invertibility during the diffusion process, we introduce a small regularization term $\delta\mathbf{I}$. Using the incidence matrix $\mathbf{B} \in \mathbb{R}^{M \times N}$ and capacity matrix $\mathbf{C} = \text{diag}(\mathbf{c})$, we define the regularized structural Laplacian $\mathbf{L}_{\text{flow}}$ as:

$$\mathbf{L}_{\text{flow}} = \mathbf{B}^\top \mathbf{C}\mathbf{B} + \delta\mathbf{I} = \sum_{(i,j) \in \mathbf{A}_{\text{sc}}} c_{ij}(\mathbf{e}_i - \mathbf{e}_j)(\mathbf{e}_i - \mathbf{e}_j)^\top + \delta\mathbf{I}. \tag{7}$$

Here, $\mathbf{L}_{\text{flow}}$ is symmetric positive definite, characterizing the diffusion properties of the brain's structural skeleton.

## 4.2. Differentiable Information Flow Solver

In this section, drawing inspiration from network flow theory and diffusion dynamics, we formulate a closed-form solution to estimate the latent communication traffic, which is the theoretical core of AFR-Net. Specifically, this traffic represents the information flow distributed across the structural connectivity, where each structural edge acts as a channel whose flow intensity is driven by global functional demands rather than just local connectivity.

### 4.2.1. DERIVING INFORMATION FLOW INTENSITY

How does information propagate through this network to support functional connectivity? We model this process by seeking a global flow equilibrium.

Consider a unit information transfer demand from a source node $s$ to a target node $t$. This creates a "potential difference" across the network, driving information flow. Mathematically, the induced state distribution $\mathbf{v}^{(st)}$ across all nodes implies solving the linear system:

$$\mathbf{L}_{\text{flow}}\mathbf{v}^{(st)} = (\mathbf{e}_s - \mathbf{e}_t) \implies \mathbf{v}^{(st)} = \mathbf{L}_{\text{flow}}^{-1}(\mathbf{e}_s - \mathbf{e}_t). \tag{8}$$

This formulation, inspired by Kirchhoff's laws, captures the global dependency of the network: a change in capacity at one edge affects the flow distribution everywhere.

For a specific edge $e_{ij} \in \mathbf{A}_{\text{sc}}$, we define the information flow intensity $I_{ij}^{(st)}$ for this pair $(s, t)$ as:

$$\begin{aligned} I_{ij}^{(st)} &= c_{ij} \left[ (\mathbf{e}_i - \mathbf{e}_j)^\top \mathbf{v}^{(st)} \right]^2 \\ &= c_{ij} \left[ (\mathbf{e}_i - \mathbf{e}_j)^\top \mathbf{L}_{\text{flow}}^{-1}(\mathbf{e}_s - \mathbf{e}_t) \right]^2. \end{aligned} \tag{9}$$

### 4.2.2. AGGREGATING GLOBAL COMMUNICATION PATTERNS

The brain operates as a massive network of concurrent interactions. The Functional Connectivity matrix $\mathbf{A}_{\text{fc}} \in \mathbb{R}^{N \times N}$ captures the statistical dependencies (e.g., Pearson correlation) between the time-series activities of brain regions. To quantify the magnitude of information exchange required to support these dependencies, we define the communication demand between nodes $s$ and $t$ as the absolute strength of

their functional connectivity $|(\mathbf{A}_{\text{fc}})_{st}|$. We use the absolute value deliberately: $|(\mathbf{A}_{\text{fc}})_{st}|$ models the *magnitude* of communication demand, since both positive correlations and anti-correlations indicate a statistical dependency that must be supported by information transmission through the shared structural substrate. This is consistent with the network communication literature, where the strength (rather than the sign) of functional coupling is used to gauge how much a structural channel is recruited (Seguin et al., 2020; 2023). Modeling signed (excitatory vs. inhibitory) demand would require a directed/asymmetric formulation and is left for future work (Section 7).

Then, the total information flow $\Phi_{ij}$ on edge $e_{ij}$ is formulated as the aggregated flow across all functional pairs, weighted by their demand:

$$\begin{aligned} \Phi_{ij} &= \sum_{(s,t) \in \mathbf{A}_{\text{fc}}} |(\mathbf{A}_{\text{fc}})_{st}| \cdot I_{ij}^{(st)} \\ &= 2c_{ij}(\mathbf{e}_i - \mathbf{e}_j)^\top \left( \mathbf{L}_{\text{flow}}^{-1} \mathbf{L}_{\text{fc}} \mathbf{L}_{\text{flow}}^{-1} \right) (\mathbf{e}_i - \mathbf{e}_j), \end{aligned} \tag{10}$$

where $\mathbf{L}_{\text{fc}} = \mathbf{D}_{\text{fc}} - |\mathbf{A}_{\text{fc}}|$. This formula elegantly unifies the local capacity ($c_{ij}$), global structural topology ($\mathbf{L}_{\text{flow}}^{-1}$), and functional demands ($\mathbf{L}_{\text{fc}}$). A high $\Phi_{ij}$ identifies edge $e_{ij}$ as a critical pathway—highlighting both direct highways and essential structural detours that are recruited to optimize information transfer.

## 4.3. Pattern-Guided Aggregation

To utilize the discovered routing map, we convert the raw flow $\mathbf{\Phi}$ into a soft mask $\mathbf{M}$ using a differentiable Log-Min-Max normalization:

$$\mathbf{M} = \sigma\left(\tau \cdot (\text{Norm}(\log(\mathbf{\Phi} + \epsilon)) - \theta)\right), \tag{11}$$

where $\epsilon = 10^{-6}$ prevents singularity, $\sigma(\cdot)$ is the Sigmoid function, and $\tau, \theta$ are learnable scaling and threshold parameters. Finally, we employ a masked transformer. Unlike standard attention, we bias the self-attention map with the learned routing mask $\mathbf{M}$:

$$\mathbf{Z}' = \text{Attention}(\mathbf{QH}, \mathbf{KH}, \mathbf{VH}, \text{bias} = \mathbf{M}). \tag{12}$$

This ensures that message passing is strictly guided by the latent communication patterns discovered by AFR-Net. The graph-level representation is pooled and fed to a classifier trained with standard Cross-Entropy loss.

**Optimization.** Let $\mathbf{z}_{\mathcal{G}}^{(k)}$ denote the pooled representation of the $k$-th subject. The final diagnosis probability is obtained via $\hat{\mathbf{y}}^{(k)} = \text{Softmax}(\text{MLP}(\mathbf{z}_{\mathcal{G}}^{(k)}))$. The entire AFR-Net framework is trained end-to-end by minimizing the standard Cross-Entropy loss:

$$\mathcal{L} = -\frac{1}{K} \sum_{k=1}^{K} \sum_{c=1}^{C} \mathbb{I}(y^{(k)} = c) \log \hat{y}_c^{(k)}, \tag{13}$$

*Table 1.* Performance comparison of different methods on ABCD and PPMI datasets (mean $\pm$ std %). Due to space limitations, only F1 and AUC scores are reported here. The complete results are available in the Appendix G. The best results are highlighted in **bold**, and the second-best results are underlined.

| Method | ABCD-OCD | | ABCD-ADHD | | ABCD-Anx | | PPMI | |
|---|---|---|---|---|---|---|---|---|
| | F1 | AUC | F1 | AUC | F1 | AUC | F1 | AUC |
| MLP | $60.48 \pm 2.27$ | $68.62 \pm 0.88$ | $61.75 \pm 3.17$ | $57.55 \pm 5.06$ | $53.03 \pm 2.44$ | $56.83 \pm 0.65$ | $54.00 \pm 9.10$ | $70.52 \pm 0.79$ |
| GCN | $52.26 \pm 10.45$ | $47.11 \pm 3.13$ | $51.28 \pm 21.88$ | $50.81 \pm 9.23$ | $52.47 \pm 8.95$ | $45.02 \pm 3.33$ | $68.79 \pm 1.37$ | $64.11 \pm 5.57$ |
| GAT | $59.90 \pm 2.18$ | $63.58 \pm 1.10$ | $59.59 \pm 2.77$ | $59.31 \pm 1.37$ | $54.95 \pm 3.45$ | $54.37 \pm 1.51$ | $70.19 \pm 0.88$ | $70.83 \pm 4.25$ |
| GIN | $42.93 \pm 11.59$ | $46.97 \pm 6.20$ | $58.31 \pm 5.85$ | $44.37 \pm 4.37$ | $49.50 \pm 10.60$ | $48.01 \pm 0.78$ | $71.43 \pm 0.00$ | $50.00 \pm 0.00$ |
| GraphSAGE | $59.76 \pm 2.12$ | $65.81 \pm 2.12$ | $47.81 \pm 5.84$ | $52.60 \pm 3.32$ | $53.75 \pm 0.54$ | $56.15 \pm 1.84$ | $73.76 \pm 4.04$ | $70.31 \pm 7.15$ |
| Graphormer | $59.40 \pm 4.73$ | $68.01 \pm 1.19$ | $55.19 \pm 5.09$ | $58.79 \pm 2.17$ | $51.34 \pm 0.87$ | $51.71 \pm 0.27$ | $72.49 \pm 1.55$ | $61.77 \pm 5.02$ |
| Triplet | $43.52 \pm 4.14$ | $52.56 \pm 0.03$ | $47.91 \pm 5.07$ | $48.84 \pm 3.72$ | $39.56 \pm 7.83$ | $48.07 \pm 0.91$ | $53.17 \pm 25.82$ | $66.25 \pm 9.51$ |
| AGT | $51.52 \pm 5.47$ | $54.18 \pm 3.17$ | $41.91 \pm 7.66$ | $53.55 \pm 4.10$ | $38.54 \pm 9.82$ | $52.51 \pm 1.10$ | $64.29 \pm 10.10$ | $61.20 \pm 1.69$ |
| BQN | $53.29 \pm 18.91$ | $57.29 \pm 3.26$ | $53.41 \pm 18.75$ | $57.11 \pm 0.68$ | $53.35 \pm 18.84$ | $51.30 \pm 0.69$ | $56.36 \pm 18.10$ | $60.73 \pm 13.60$ |
| BrainGNN | $54.82 \pm 2.79$ | $53.53 \pm 0.10$ | $51.35 \pm 20.84$ | $54.37 \pm 5.89$ | $47.71 \pm 3.10$ | $55.38 \pm 4.06$ | $71.02 \pm 4.62$ | $71.30 \pm 4.41$ |
| Cross-GNN | $58.21 \pm 1.75$ | $58.31 \pm 1.89$ | $52.76 \pm 3.79$ | $56.24 \pm 3.94$ | $50.08 \pm 2.05$ | $50.56 \pm 1.25$ | $63.51 \pm 5.42$ | $65.83 \pm 8.74$ |
| MaskGNN | $58.42 \pm 2.47$ | $63.30 \pm 1.12$ | $49.22 \pm 1.50$ | $61.58 \pm 2.05$ | $50.36 \pm 3.40$ | $52.67 \pm 0.72$ | $73.50 \pm 1.11$ | $68.23 \pm 0.74$ |
| RH-BrainFS | $59.55 \pm 3.15$ | $59.69 \pm 3.12$ | $61.59 \pm 2.94$ | $61.76 \pm 2.61$ | $53.69 \pm 5.26$ | $54.75 \pm 5.36$ | $68.54 \pm 3.15$ | $68.25 \pm 3.06$ |
| NeuroPath | $\underline{64.13 \pm 2.10}$ | $\underline{69.39 \pm 0.15}$ | $\underline{62.36 \pm 3.71}$ | $\mathbf{64.19 \pm 0.35}$ | $55.56 \pm 2.24$ | $57.65 \pm 2.07$ | $\underline{74.56 \pm 0.76}$ | $\underline{71.88 \pm 1.13}$ |
| Ours | $\mathbf{70.62 \pm 2.36}$ | $\mathbf{72.68 \pm 1.78}$ | $\mathbf{65.50 \pm 2.35}$ | $\underline{62.38 \pm 1.08}$ | $\mathbf{62.15 \pm 2.41}$ | $\mathbf{59.03 \pm 1.94}$ | $\mathbf{83.70 \pm 3.85}$ | $\mathbf{90.38 \pm 1.64}$ |

where $y^{(k)}$ is ground-truth and $\mathbb{I}(\cdot)$ is indicator function.

## 5. Experiments

### 5.1. Experimental Settings

**Datasets.** We comprehensively evaluate AFR-Net on two multi-modal brain network benchmarks: ABCD and PPMI. Detailed descriptions are provided in Appendix D.1.

**Evaluation Metrics.** To provide a holistic performance assessment, we utilize five standard metrics: Accuracy (Acc), Precision (Pre), Recall (Rec), F1-score (F1), and Area Under the ROC Curve (AUC). These metrics cover different aspects of classification performance, ensuring robustness against class imbalance. The precise mathematical formulations for all metrics are detailed in Appendix D.3.

**Baselines.** We compare AFR-Net against a comprehensive set of 14 baselines (6 general graph-learning backbones and 8 specialized brain-network models). **All baselines receive both SC and FC.** For the general graph-learning methods, which were not designed for multimodal data, we use a *dual-stream* adaptation: SC and FC are each encoded by a separate same-type encoder and the resulting embeddings are concatenated before the classification head, ensuring equal access to both modalities. For the specialized multimodal methods, we follow each method's original design for ingesting SC and FC. We deliberately retain the general methods because their comparatively weaker performance shows that naive dual-stream fusion cannot capture the complex SC–FC coupling, motivating mechanism-driven designs such as AFR-Net. The two groups are:

**General Graph Learning Methods:** We include stan-

dard backbones such as MLP (Rumelhart et al., 1986), GCN (Kipf & Welling, 2017), GAT (Velickovic et al., 2018), GraphSAGE (Hamilton et al., 2017), GIN (Xu et al., 2019), and Graphormer (Ying et al., 2021).

**State-of-the-Art Brain Network Methods:** We compare against specialized models designed for brain connectivity analysis, including BrainGNN (Li et al., 2021), Triplet (Zhu et al., 2022), BQN (Yang et al., 2025), AGT (Cho et al., 2024), Cross-GNN (Yang et al., 2024), MaskGNN (Qu et al., 2025), RH-BrainFS (Ye et al., 2023), and NeuroPath (Wei et al., 2024b). Due to space limitations, the description of the baselines has been placed in Appendix D.2.

### 5.2. Results and Analysis

The classification performance of AFR-Net and all baseline methods on four datasets is summarized in Table 1. From the experimental results, several key observations can be made. First, generally superior performance is observed across all models on the PPMI and OCD datasets, whereas performance on the ADHD and Anx datasets is consistently lower. This suggests that the prediction tasks for Attention-Deficit/Hyperactivity Disorder and Anxiety are more challenging, potentially due to the subtle nature of the connectivity alterations associated with these conditions.

Second, it is noteworthy that traditional GNN-based methods (e.g., GCN, GAT, GIN) and methods specifically designed for brain networks such as Cross-GNN and MaskGNN yield suboptimal performance. This corroborates the limitations of local aggregation mechanisms when applied to brain networks. In contrast, RH-BrainFS, which introduces a fusion bottleneck to capture the coupling strength between FC and SC, and NeuroPath, which accounts for the topological detour phenomenon, achieve

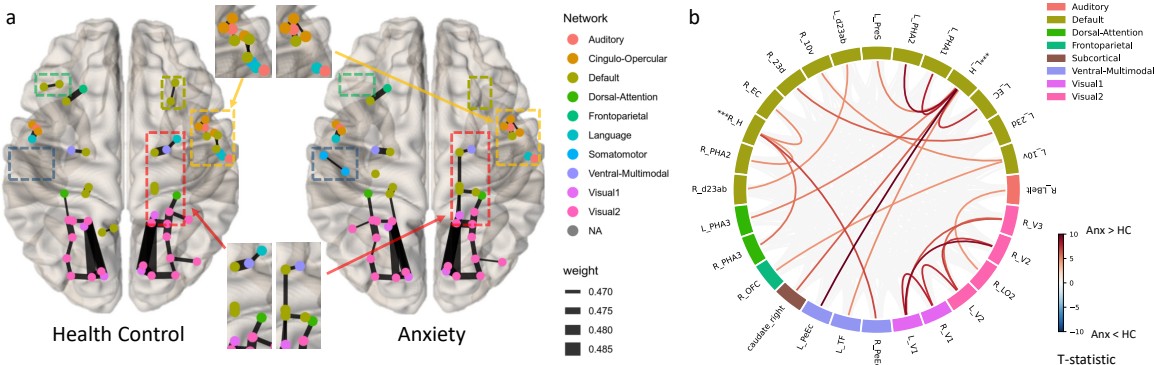

Figure 3. (a) Visualization of the top 100 edges with the highest information flow intensity ($\Phi_{ij}$) averaged across all subjects. The model autonomously identifies the visual and somatomotor networks as the structural core of brain communication. (b) Circle plot showing edges with significant group differences ($p < 0.05$, FDR corrected). Red lines denote edges where the patient exhibits higher information flow intensity than HC, whereas blue lines indicate the opposite. The left hippocampus (**L_H**, marked with ***) emerges as a pathological hub, exhibiting dense hyper-connectivity with the default mode network and subcortical regions in patient group.

better results. These methods effectively address the issue of regional heterogeneity, thereby learning more expressive connectomic representations.

Crucially, our proposed AFR-Net achieves the best overall performance, ranking first on the large majority of metrics across all four tasks. The only exception is AUC on the most challenging ABCD-ADHD task, where AFR-Net is second to NeuroPath (62.38 vs. 64.19); notably, AFR-Net still attains the best ADHD F1 (65.50 vs. 62.36). This reflects a recall–precision trade-off: AFR-Net obtains substantially higher recall (71.67% vs. 69.17%) at a slight precision cost, which raises F1 while marginally lowering AUC. Moreover, the 1.8-point AUC gap lies within one standard deviation, and ADHD is the hardest task where all methods operate close to chance; in clinical screening, higher recall (fewer missed cases) is generally preferable. Overall, these results demonstrate that AFR-Net effectively captures how the interactions and communications between neural elements are constrained by the topological architecture of the SC, and how these dynamic patterns of information flow give rise to the FC. This underscores the superiority of fusing FC and SC representations from a more neuroscientifically grounded perspective.

Finally, it is worth noting that MLP performs competitively among the baselines, even surpassing some GNN methods. This finding aligns with the work of Popov et al. (2024); Yang et al. (2025); Hou et al. (2025), suggesting that increasing model complexity does not necessarily lead to performance gains if the underlying brain network is not modeled accurately.

### 5.3. Interpretability Analysis

To validate that AFR-Net captures biologically meaningful signal routing, we conduct a two-stage analysis on the Anx-

iety dataset. First, we visualize the informational backbone by extracting the top 100 edges with the highest flow intensity ($\Phi_{ij}$). As shown in Figure 3(a), these high-traffic edges are predominantly concentrated within the visual and somatomotor networks. This aligns with the brain's established functional hierarchy, where sensory-motor processing constitutes the dense "structural core" of neural communication (Bullmore & Sporns, 2009), confirming that AFR-Net captures brain connectivity patterns consistent with established neuroscience, even without explicit supervision. We further verify in Appendix F that these patterns are not a trivial reflection of raw SC weights and are highly stable across random initializations.

From Figure 3(a), we can also observe disparities in connectivity patterns between the patient and healthy control (HC) groups. To quantify these differences, we perform a two-sample t-test to identify signal pathways with statistically significant deviations. As visualized in Figure 3(b), the results demonstrate that for all significantly distinct edges, the information flow intensity in patients is consistently higher than that in healthy controls. This indicates an elevated communication load along these routes, revealing a distinct pattern of hyper-active information traffic within the Limbic-DMN (default mode network, a key brain system active during self-reflection and mind-wandering) connection. Specifically, the left hippocampus (L_H) acts as a pathological hub, exhibiting heightened neural communication with the DMN and subcortical regions compared to HC. To interpret the cognitive implication of this hub, we perform functional decoding via Neurosynth (Yarkoni et al., 2011). The results indicate that the L_H region is most strongly associated with cognitive terms such as "autobiographical", "episodic", and "retrieval". Synthesizing our connectivity findings with this functional evidence offers a mechanistic explanation for anxiety. Existing neuroscience

*Table 2.* Ablation study of the main modules in our proposed AFR-Net method. The best results are highlighted in bold.

| Dataset | Metric | w/o ERD | w/o Flow Routing | Full Model |
|---------|--------|---------|------------------|------------|
| OCD | F1 | $67.74 \pm 2.25$ | $60.53 \pm 6.23$ | $\mathbf{70.62 \pm 2.36}$ |
|     | AUC | $71.73 \pm 2.34$ | $68.21 \pm 1.38$ | $\mathbf{72.68 \pm 1.78}$ |
| ADHD | F1 | $63.70 \pm 2.30$ | $60.12 \pm 2.57$ | $\mathbf{65.50 \pm 2.35}$ |
|      | AUC | $60.90 \pm 1.80$ | $59.23 \pm 1.75$ | $\mathbf{62.38 \pm 1.08}$ |
| Anx | F1 | $52.95 \pm 8.29$ | $49.70 \pm 4.48$ | $\mathbf{62.15 \pm 2.41}$ |
|     | AUC | $58.95 \pm 1.46$ | $58.00 \pm 2.63$ | $\mathbf{59.03 \pm 1.94}$ |

literature posits that anxiety disorders are characterized by pathological rumination—the repetitive rehearsal of negative past events (Hamilton et al., 2015). While the DMN supports self-referential processing, the hippocampus is critical for episodic memory retrieval. The hyper-active flow we observed from the Hippocampus to the DMN strongly suggests a maladaptive circuit where intrusive autobiographical memories are relentlessly retrieved and projected into self-referential loops. This finding is consistent with the triple network model of psychopathology (Menon, 2011), which implicates dysregulated switching between memory and self-processing circuits in internalizing disorders. By uncovering this specific circuit, AFR-Net demonstrates its value in generating testable, mechanically grounded neuroscientific hypotheses.

### 5.4. Ablation Study

To validate the contribution of each component in AFR-Net, we conducted an ablation study on the three ABCD datasets. We compared the full model against two variants: (1) *w/o ERD*, which replaces the effective resistance distance with standard positional encodings; and (2) *w/o Flow Routing*, which removes the adaptive flow layer and relies solely on the Transformer encoder for classification. The results are summarized in Table 2.

The removal of the flow routing module results in the most significant performance degradation across all tasks. Notably, on the Anxiety dataset, the F1-score drops precipitously from $62.15\%$ to $49.70\%$, and in OCD, it falls by over $10\%$. This sharp decline underscores that the flow routing module is indispensable as it dynamically simulates the global energy equilibrium, allowing the model to uncover latent communication pathways driven by functional demands that are otherwise invisible to standard topological aggregators. The variant without ERD encoding also exhibits a consistent performance drop compared to the full model. This verifies our hypothesis that brain communication relies on parallel redundancy rather than just shortest paths. By encoding ERD, the model gains a global view of structural robustness from the initialization stage.

To isolate the circuit-theoretic inductive bias from the learned edge gate, and to further assess statistical and inter-

pretational robustness, we additionally conduct a raw-SC-capacity ablation, 5-fold cross-validation on the small PPMI cohort, and SC-based null-model and cross-seed consistency checks. All of these analyses corroborate our conclusions and, due to space constraints, are reported in detail in Appendix F.

## 6. Conclusion

We reconceptualize the coupling of structural and functional connectivity not as a static mapping, but as a dynamic, energy-optimal flow process. By integrating physics-informed graph construction with a differentiable flow solver, AFR-Net uncovers the critical pathways underlying neural communication. Beyond achieving state-of-the-art performance across large-scale benchmarks, AFR-Net bridges the chasm between predictive power and biological interpretability. By elucidating the specific routing patterns driving pathological alterations, our framework transforms deep learning from a black-box predictor into a mechanistic instrument for neuroscientific discovery.

## 7. Limitations

AFR-Net has four main limitations, each pointing to future work. **(i) Scaling:** the Cholesky factorization of $\mathbf{L}_{\text{flow}}$ is $\mathcal{O}(N^3)$—efficient for region-level atlases ($N=90$–$400$; Appendix C), as used by all our baselines, but a ceiling for voxel-level connectomes ($N>10{,}000$); sparse/Nyström solvers or graph coarsening could relax it while preserving the Kirchhoff formulation. **(ii) Static demand:** we use a static FC matrix, though the framework extends to windowed $\mathbf{L}_{\text{fc}}(t)$ for dynamic FC. **(iii) Undirected flow:** since fMRI/DTI are undirected, our flow captures traffic *magnitude* rather than direction; an asymmetric demand term would add directionality without changing the solver. **(iv) Atlas and sample size:** gains are consistent across three atlases (HCP-MMP1.0, FreeSurfer, AAL), and although PPMI is small it is cross-validated (Appendix F) with the large ABCD cohort providing the primary evidence, but prospective multi-site validation is still needed for clinical use.

## Acknowledgements

The authors would like to thank the anonymous reviewers for their constructive comments. This work was supported in part by the Commonwealth Cyber Initiative (CCI) under Award No. VV-1Q26-005 and the National Science Foundation under Grant No. 2331315. Any opinions, findings, and conclusions or recommendations expressed in this material are those of the authors and do not necessarily reflect the views of the National Science Foundation.

## Impact Statement

Our research aims to enhance the interpretability and accuracy of brain network analysis for medical diagnosis. By explicitly modeling neural information flow, AFR-Net offers a more transparent mechanism for identifying pathological routing patterns compared to black-box deep learning approaches. This improved interpretability is crucial for building trust in AI-assisted healthcare. We emphasize that any clinical application of this technology requires rigorous validation to ensure safety and fairness. We do not foresee any immediate negative societal consequences, provided that the technology is used ethically and under human supervision.

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

## A. Summary of Notations

Here, we summarize the notations used throughout the paper in Table 3. Table 4 further clarifies which quantities are fixed by the data versus learned end-to-end, showing that the circuit formulation acts as a structural inductive bias while only a small set of components is optimized.

*Table 3.* Summary of notations used in this paper.

| Symbol | Description |
|---|---|
| $\mathcal{G}_{sc}, \mathcal{G}_{fc}$ | Structural and Functional brain network graphs |
| $N, M$ | Number of nodes (ROIs) and structural edges |
| $\mathbf{A}_{sc}, \mathbf{A}_{fc}$ | Adjacency matrices for SC and FC |
| $\mathbf{X}, \mathbf{H}$ | Input node features and encoded hidden representations |
| $\mathbf{B}$ | Node-edge incidence matrix $\in \mathbb{R}^{M \times N}$ |
| $R_{ij}$ | Effective Resistance Distance (ERD) between nodes $v_i$ and $v_j$ |
| $c_{ij}, \mathbf{C}$ | Learned flow capacity of edge $e_{ij}$ and its diagonal matrix |
| $\mathbf{L}_{flow}$ | Regularized structural Laplacian ($\mathbf{B}^\top \mathbf{CB} + \delta \mathbf{I}$) |
| $\mathbf{L}_{fc}$ | Functional demand Laplacian ($\mathbf{D}_{fc} - |\mathbf{A}_{fc}|$) |
| $\mathbf{v}^{(st)}$ | Node potential vector induced by unit demand from $s$ to $t$ |
| $I_{ij}^{(st)}$ | Information flow intensity on edge $e_{ij}$ for node pair $(s, t)$ |
| $\Phi_{ij}, \mathbf{\Phi}$ | Aggregated information flow on edge $e_{ij}$ and the flow map |
| $\mathbf{M}$ | Pattern-guided soft attention mask (routing mask) |
| $y, \hat{y}$ | Ground-truth label and predicted probability |

## B. Theoretical Proofs

In this section, we provide the detailed mathematical derivation for the closed-form solution of the total information flow presented in Eq. (10).

**Theorem B.1** (Closed-Form Solution of Total Information Flow). *Given a structural flow network characterized by the regularized Laplacian $\mathbf{L}_{flow} = \mathbf{B}^\top \mathbf{CB} + \delta \mathbf{I}$ and a functional demand matrix $|\mathbf{A}_{fc}|$ with its corresponding Laplacian $\mathbf{L}_{fc}$. Let $c_{ij}$ be the capacity of edge $e_{ij} \in \mathbf{A}_{sc}$. The total expected information flow $\Phi_{ij}$, aggregated over all communicating pairs $(s, t)$ weighted by demand $|(\mathbf{A}_{fc})_{st}|$, is given by:*

$$\Phi_{ij} = 2c_{ij}(\mathbf{e}_i - \mathbf{e}_j)^\top \left( \mathbf{L}_{flow}^{-1} \mathbf{L}_{fc} \mathbf{L}_{flow}^{-1} \right) (\mathbf{e}_i - \mathbf{e}_j). \tag{14}$$

*Proof.* Recall the definition of the information flow intensity $I_{ij}^{(st)}$ induced on edge $e_{ij}$ caused by a unit flow demand from source $s$ to target $t$. Based on the global flow equilibrium (analogous to Ohm's law on graphs (Ohm, 1827)), the potential distribution is given by the linear system $\mathbf{L}_{flow}\mathbf{v}^{(st)} = (\mathbf{e}_s - \mathbf{e}_t)$. Since $\mathbf{L}_{flow}$ is regularized by $\delta\mathbf{I}$, it is strictly positive definite and invertible, yielding the unique solution $\mathbf{v}^{(st)} = \mathbf{L}_{flow}^{-1}(\mathbf{e}_s - \mathbf{e}_t)$. Note that as $\delta \to 0$, this approaches the pseudoinverse solution $\mathbf{L}^\dagger(\mathbf{e}_s - \mathbf{e}_t)$ for connected components where $\mathbf{1}^\top(\mathbf{e}_s - \mathbf{e}_t) = 0$.

The flow intensity is defined as capacity times the squared potential difference:

$$I_{ij}^{(st)} = c_{ij} \left[ (\mathbf{e}_i - \mathbf{e}_j)^\top \mathbf{L}_{flow}^{-1}(\mathbf{e}_s - \mathbf{e}_t) \right]^2. \tag{15}$$

The total information flow $\Phi_{ij}$ is the weighted sum of microscopic intensities over all possible pairs:

$$\Phi_{ij} = \sum_{s=1}^{N} \sum_{t=1}^{N} |(\mathbf{A}_{fc})_{st}| \cdot I_{ij}^{(st)} \tag{16}$$

*Table 4.* Which quantities in AFR-Net are fixed by the data versus learned end-to-end. This clarifies that the circuit formulation acts as a structural inductive bias, while only a small set of components is optimized.

| Category | Quantity | Description |
|---|---|---|
| Fixed (data) | $\mathbf{A}_{\text{sc}}, \mathbf{A}_{\text{fc}}$ | Input SC / FC connectivity matrices |
| Fixed (data) | $\mathbf{B}$ | Incidence matrix (from $\mathbf{A}_{\text{sc}}$ topology) |
| Precomputed | $R_{ij}$ | ERD from $\mathbf{L}_{\text{sc}}^{\dagger}$ |
| Precomputed | $\mathbf{L}_{\text{fc}}$ | Functional demand Laplacian from $|\mathbf{A}_{\text{fc}}|$ |
| Learned | $\mathbf{Q}, \mathbf{K}, \mathbf{V}$ | Transformer projections (per layer) |
| Learned | $\psi_{\text{res}}(\cdot)$ | ERD encoder (2-layer MLP, 128, GELU) |
| Learned | $\omega_{\text{gate}}(\cdot) \to c_{ij}$ | Edge-gating net (2-layer MLP, 64, SiLU) |
| Learned | $\tau, \theta$ | Mask scaling / threshold (Eq. 11) |
| Learned | MLP head | Classifier |

Let $\mathbf{w}_{ij} = \mathbf{L}_{\text{flow}}^{-1}(\mathbf{e}_i - \mathbf{e}_j)$ denote the structural response vector specific to edge $e_{ij}$. The term inside the square bracket becomes $\mathbf{w}_{ij}^{\top}(\mathbf{e}_s - \mathbf{e}_t)$. Using the trace property of scalar values ($x^2 = \text{Tr}(xx^{\top})$), we rewrite $I_{ij}^{(st)}$:

$$I_{ij}^{(st)} = c_{ij}\text{Tr}\left(\mathbf{w}_{ij}^{\top}(\mathbf{e}_s - \mathbf{e}_t)(\mathbf{e}_s - \mathbf{e}_t)^{\top}\mathbf{w}_{ij}\right) = c_{ij}\mathbf{w}_{ij}^{\top}\left[(\mathbf{e}_s - \mathbf{e}_t)(\mathbf{e}_s - \mathbf{e}_t)^{\top}\right]\mathbf{w}_{ij}. \tag{17}$$

Substituting this back into the summation for $\Phi_{ij}$:

$$\Phi_{ij} = c_{ij}\mathbf{w}_{ij}^{\top}\left[\sum_{s,t\in\mathbf{A}_{\text{fc}}} |(\mathbf{A}_{\text{fc}})_{st}|(\mathbf{e}_s - \mathbf{e}_t)(\mathbf{e}_s - \mathbf{e}_t)^{\top}\right]\mathbf{w}_{ij}. \tag{18}$$

We now analyze the term inside the bracket. Let $\mathbf{Q} = \sum_{s,t} |(\mathbf{A}_{\text{fc}})_{st}|(\mathbf{e}_s - \mathbf{e}_t)(\mathbf{e}_s - \mathbf{e}_t)^{\top}$. We expand the outer product:

$$(\mathbf{e}_s - \mathbf{e}_t)(\mathbf{e}_s - \mathbf{e}_t)^{\top} = \mathbf{e}_s\mathbf{e}_s^{\top} - \mathbf{e}_s\mathbf{e}_t^{\top} - \mathbf{e}_t\mathbf{e}_s^{\top} + \mathbf{e}_t\mathbf{e}_t^{\top}. \tag{19}$$

The matrix $\mathbf{Q}$ corresponds to the definition of the graph Laplacian for a graph with adjacency matrix $|\mathbf{A}_{\text{fc}}|$. Specifically, the $(i,j)$-th entry of $\mathbf{Q}$ (where $i \neq j$) collects terms $-(\mathbf{A}_{\text{fc}})_{ij}$ and $-(\mathbf{A}_{\text{fc}})_{ji}$. Since $\mathbf{A}_{\text{fc}}$ is symmetric ($(\mathbf{A}_{\text{fc}})_{ij} = (\mathbf{A}_{\text{fc}})_{ji}$), the off-diagonal entry is $-2|(\mathbf{A}_{\text{fc}})_{ij}|$. The diagonal entry $Q_{ii}$ sums up degrees. Thus, we identify:

$$\mathbf{Q} = 2\mathbf{L}_{\text{fc}}, \tag{20}$$

where $\mathbf{L}_{\text{fc}}$ is the Laplacian of the functional demand graph.

Finally, substituting $\mathbf{Q} = 2\mathbf{L}_{\text{fc}}$ and the definition of $\mathbf{w}_{ij}$ back into Eq. (18):

$$\Phi_{ij} = c_{ij}\mathbf{w}_{ij}^{\top}(2\mathbf{L}_{\text{fc}})\mathbf{w}_{ij} = 2c_{ij}(\mathbf{e}_i - \mathbf{e}_j)^{\top}\mathbf{L}_{\text{flow}}^{-1}\mathbf{L}_{\text{fc}}\mathbf{L}_{\text{flow}}^{-1}(\mathbf{e}_i - \mathbf{e}_j). \tag{21}$$

$\square$

## C. Computational Complexity Analysis

While AFR-Net incorporates a global flow computation involving the inversion of the Laplacian matrix, we ensure high computational efficiency through algorithmic optimizations and the intrinsic properties of brain networks.

**Theoretical Analysis.** The computational bottleneck of our routing module lies in evaluating the quadratic form $\mathbf{x}^{\top}\mathbf{L}_{\text{flow}}^{-1}\mathbf{L}_{\text{fc}}\mathbf{L}_{\text{flow}}^{-1}\mathbf{x}$. A naive inversion of $\mathbf{L}_{\text{flow}}$ would incur a complexity of $\mathcal{O}(N^3)$. However, we leverage the positive-definiteness of the regularized Laplacian $\mathbf{L}_{\text{flow}} = \mathbf{B}^{\top}\mathbf{C}\mathbf{B} + \delta\mathbf{I}$ to employ the Cholesky decomposition $\mathbf{L}_{\text{flow}} = \mathbf{L}\mathbf{L}^{\top}$.

- Forward Pass (Amortized Cost): The Cholesky factorization is computed once per graph with a complexity of $\frac{1}{3}N^3$. Once factorized, solving the linear system $\mathbf{L}_{\text{flow}}\mathbf{z} = \mathbf{x}$ for latent representations reduces to forward and backward substitutions, costing $\mathcal{O}(N^2 d)$.

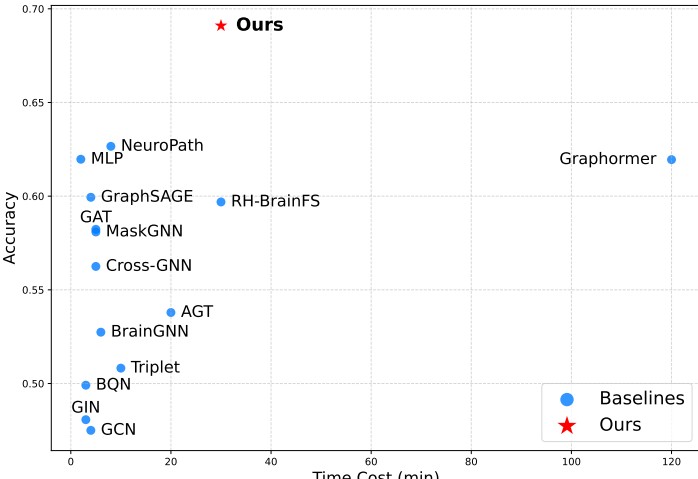

*Figure 4.* Accuracy vs. Time Cost on ABCD dataset over 50 epochs for each method. Our method achieves the best trade-off between performance and efficiency.

- Scale Analysis: In the context of brain network analysis, the number of nodes $N$ corresponds to Regions of Interest (ROIs), which typically ranges from 100 to 400 (e.g., AAL or Schaefer atlases). In contrast, the hidden dimension $d$ in Transformers is often large (e.g., 64-512). Consequently, the cubic term $N^3$ is comparable in magnitude to the quadratic term $N^2 d$ dominant in the self-attention mechanism. Modern GPUs are highly optimized for such dense matrix operations, rendering the overhead of the flow module manageable.

- Backward Pass (Implicit Differentiation): Crucially, we do not need to backpropagate through the unrolled steps of the Cholesky factorization or the linear solver. Instead, we utilize implicit differentiation via the adjoint method. The gradient $\partial \mathcal{L}/\partial \mathbf{L}_{\text{flow}}$ can be computed by solving a symmetric linear system of the same scale, maintaining the same $\mathcal{O}(N^3)$ complexity class as the forward pass while significantly reducing memory consumption.

**Comparison with Baselines.** The standard Graphormer encoder has a complexity of $\mathcal{O}(L \cdot N^2 \cdot d + L \cdot N^2 \cdot K)$, where $L$ is the number of layers and $K$ is the path length for edge encoding. Since AFR-Net performs the flow computation only once (or once per layer depending on architecture) compared to the multi-head attention computed $L \times H$ times, the relative overhead is minimal.

We also report the running times of all methods on the ABCD-OCD dataset over 50 epochs in Figure 4. As shown, our method exhibits comparable computational efficiency to RH-BrainFS, with runtime that remains well within practical limits. To conclude, AFR-Net achieves the best trade-off between accuracy and training time.

## D. Details of Experimental Setup

### D.1. Datasets

We evaluate AFR-Net on two large-scale multi-modal neuroimaging benchmarks: the Adolescent Brain Cognitive Development study and the Parkinson's Progression Markers Initiative.

**Adolescent Brain Cognitive Development (ABCD).** The ABCD study is the largest long-term prospective longitudinal study of brain development and child health in the United States. It recruited a representative sample of 11,875 children aged 9-10 years from 21 research sites across the country. In this work, we utilized baseline data from Data Release 6.0. We applied rigorous inclusion criteria, excluding participants with missing neuroimaging (rs-fMRI or d-MRI) or behavioral data, as well as those flagged for poor data quality, resulting in a final cohort of 6,381 participants. For image processing, we utilized rs-fMRI data processed according to the standardized ABCD pipeline, which includes motion correction, $B_0$ distortion correction, and gradient nonlinearity distortion correction. Motion confounds were minimized using iterative smoothing, regression of motion parameters, and frame censoring. The d-MRI preprocessing was implemented using MRtrix3, involving MP-PCA denoising, reverse phase encoding, and corrections for susceptibility, eddy currents, and motion. We employed the Dhollander algorithm to estimate tissue response functions (CSF, gray matter, white matter) and

computed Fiber Orientation Distributions via spherical deconvolution. Structural connectivity was generated via probabilistic tractography (iFOD2 algorithm) with 10 million streamlines, followed by SIFT2 filtering to reconstruct biologically accurate streamline weights. Finally, consistent with our framework's requirements, we used the HCP-MMP1.0 (Glasser) and FreeSurfer (Aseg) atlases to parcellate the brain, extracting the functional connectivity matrix from time-series correlations and the structural connectivity matrix from streamline counts.

**Parkinson's Progression Markers Initiative (PPMI).** The PPMI is a landmark observational clinical study aimed at identifying biomarkers for Parkinson's Disease progression. We selected a subset of subjects who possessed both Diffusion Tensor Imaging (DTI) and Resting-State fMRI (rs-fMRI) data at the baseline visit, resulting in a cohort of 90 subjects classified into PD patients and Healthy Controls (HC). The rs-fMRI data underwent standard preprocessing using FSL and AFNI pipelines, including slice timing correction, rigid-body head motion correction, spatial normalization to MNI152 space, Gaussian smoothing, and band-pass filtering. Nuisance signals (motion parameters, CSF, and white matter signals) were regressed out. For d-MRI, raw data were corrected for eddy currents and head motion using FSL's eddy tool. Deterministic tractography was performed using the Fiber Assignment by Continuous Tracking algorithm to map white matter tracts. For feature extraction, we utilized the Automated Anatomical Labeling (AAL) atlas to parcellate the brain into 90 regions of interest. Based on this parcellation scheme, the functional connectivity matrix was derived from the Pearson correlation of the ROI-averaged time series, and the structural connectivity matrix was constructed by counting the number of fiber tracts connecting each pair of ROIs.

### D.2. Baseline Models

We compare AFR-Net against $14$ competitive baselines, categorized into $6$ general graph-learning methods (dual-stream adapted so that each receives both SC and FC) and $8$ specialized multimodal brain-network analysis methods (which ingest SC and FC following their original designs).

- **Multi-Layer Perceptron (MLP) (Rumelhart et al., 1986):** A standard deep learning baseline that flattens the upper triangular part of the connectivity matrices into a vector and feeds it into fully connected layers, ignoring the graph topological structure.

- **Graph Convolutional Network (GCN) (Kipf & Welling, 2017):** GCN is a seminal spectral-based graph neural network that extends the convolution operation to graph-structured data. It updates node representations by aggregating features from the immediate neighborhood using a fixed, isotropic normalization rule based on the degree matrix.

- **Graph Attention Network (GAT) (Velickovic et al., 2018):** GAT introduces a self-attention mechanism to learn the importance of neighbors dynamically. It computes anisotropic attention coefficients between connected brain regions, allowing the model to focus on more relevant connections during the message-passing phase.

- **GraphSAGE (Hamilton et al., 2017):** This method provides a general inductive framework that generates node embeddings by sampling and aggregating features from a node's local neighborhood. Instead of training individual embeddings for each node, GraphSAGE learns a function that aggregates feature information from local neighbors.

- **Graph Isomorphism Network (GIN) (Xu et al., 2019):** GIN is designed to maximize the expressive power of GNNs, theoretically achieving discriminative capability equivalent to the Weisfeiler-Lehman (WL) graph isomorphism test. It employs a summation aggregator followed by a Multi-Layer Perceptron to ensure that distinct graph structures map to different embeddings, making it highly effective for graph-level classification tasks in brain network analysis.

- **Graphormer (Ying et al., 2021):** As a representative of Graph Transformers, Graphormer adapts the standard Transformer architecture to graph data. It incorporates centrality encoding and spatial encoding to effectively capture both the structural information and long-range dependencies between brain regions. This allows the model to surpass the limitation of the receptive field in conventional message-passing GNNs.

- **Triplet (Zhu et al., 2022):** It employs a triplet attention network architecture to learn discriminative high-order representations and relationships among samples, utilizing attention mechanisms to fuse functional and structural features.

- **AGT (Cho et al., 2024):** AGT proposes an adaptive graph diffusion framework designed to capture diverse connectivity patterns in brain networks by learning node-wise spectral scales. It introduces a novel group-level temporal regularization which explicitly constraints the centroids of diagnostic groups in latent space to follow their natural progressive order, thereby effectively encoding the irreversible temporal dynamics of neurodegeneration into downstream task.

- **BQN (Yang et al., 2025)** BQN challenges the necessity of the conventional message-passing mechanism in brain network modeling. It employs a Quadratic Network architecture based on the Hadamard product rather than matrix multiplication which allows it to capture high-order nonlinear interactions and implicitly perform community detection via non-negative matrix factorization logic, achieving superior computational efficiency and performance without explicit message passing.

- **BrainGNN (Li et al., 2021):** BrainGNN introduces ROI-aware Graph Convolutional layers, which assign distinct parameters to each ROI to characterize their specific functional patterns. Furthermore, it incorporates a differentiable ROI-selection pooling layer to retain salient nodes with high projection scores while pruning irrelevant ones.

- **Cross-GNN (Yang et al., 2024):** Cross-GNN introduces a dynamic graph learning mechanism where an inter-modal correspondence matrix is learned to model the intrinsic dependencies between fMRI and DTI features, serving as a dynamic adjacency matrix. Furthermore, it incorporates a cross-distillation strategy to regularize latent representations via mutual learning between mono-modal and multi-modal pathways, enhancing the model's generalization ability.

- **MaskGNN (Qu et al., 2025):** MaskGNN employs a learnable edge mask mechanism to dynamically weight the importance of neural connections within a unified graph structure constructed via the Glasser atlas. Furthermore, it incorporates manifold regularization and orthonormality constraints to preserve local topological structures and ensure the learning of stable, independent features.

- **RH-BrainFS (Ye et al., 2023):** RH-BrainFS addresses the "regional heterogeneity" issue, where the coupling strength between structural and functional modalities varies across different brain regions. It employs a brain subgraph neural network to extract local regional characteristics via rooted subgraph embedding and introduces a transformer-based fusion bottleneck module that facilitates indirect interaction between modalities using learnable bottleneck tokens.

- **NeuroPath (Wei et al., 2024b):** NeuroPath introduces the concept of "topological detour" to characterize how functional interactions are physically supported by multi-hop structural pathways. The model employs a twin-branch architecture with a specialized Multi-Head Self-Attention mechanism that filters attention maps using detour adjacency matrices, enforcing consistency between structural pathways and functional correlations to learn expressive connectomic representations.

### D.3. Evaluation Metrics

To comprehensively evaluate the classification performance of our proposed AFR-Net and the baseline methods, we employ five standard metrics: Accuracy (Acc), Precision (Pre), Recall (Rec), F1-score (F1), and Area Under the ROC Curve (AUC).

Let $TP$, $TN$, $FP$, and $FN$ denote the number of True Positives, True Negatives, False Positives, and False Negatives, respectively.

**Accuracy (Acc).** This metric measures the proportion of correctly classified samples among the total number of samples:

$$\text{Acc} = \frac{TP + TN}{TP + TN + FP + FN} \tag{22}$$

**Precision (Pre).** Precision quantifies the accuracy of positive predictions, indicating the reliability of the model when it predicts a positive class:

$$\text{Pre} = \frac{TP}{TP + FP} \tag{23}$$

**Recall (Rec).** Also known as Sensitivity, Recall measures the ability of the model to identify all actual positive instances:

$$\text{Rec} = \frac{TP}{TP + FN} \tag{24}$$

*Table 5.* Robustness of PPMI results under 5-fold stratified cross-validation (3 seeds × 5 folds = 15 runs; mean ± std %). The cross-validated AFR-Net result is consistent with the original held-out split and remains substantially superior to the strongest baseline.

| Method | F1 | AUC |
|---|---|---|
| AFR-Net (5-fold CV) | $82.90 \pm 3.54$ | $89.80 \pm 1.56$ |
| AFR-Net (original split) | $83.70 \pm 3.85$ | $90.38 \pm 1.64$ |
| NeuroPath (original split) | $74.56 \pm 0.76$ | $71.88 \pm 1.13$ |

*Table 6.* Edge-capacity ablation isolating the circuit-theoretic inductive bias from the learned edge gate (F1 / AUC %). *Raw SC Capacity* replaces the learned gate with fixed raw SC weights ($c_{ij} = \mathrm{SC}_{ij}$); *w/o Flow Routing* removes the routing module entirely. Both the circuit formulation and the adaptive gating are necessary, and they are synergistic.

| Variant | OCD | ADHD | Anx |
|---|---|---|---|
| w/o Flow Routing | 60.53 / 68.21 | 60.12 / 59.23 | 49.70 / 58.00 |
| Raw SC Capacity | 57.31 / 65.64 | 55.87 / 55.03 | 55.63 / 51.12 |
| Full Model | **70.62 / 72.68** | **65.50 / 62.38** | **62.15 / 59.03** |

**F1-score (F1).** The F1-score is the harmonic mean of Precision and Recall, providing a balanced metric especially when class distributions are imbalanced:

$$\mathrm{F1} = 2 \cdot \frac{\mathrm{Pre} \cdot \mathrm{Rec}}{\mathrm{Pre} + \mathrm{Rec}} \tag{25}$$

**Area Under the Receiver Operating Characteristic Curve (AUC).** The AUC represents the probability that a randomly chosen positive instance is ranked higher than a randomly chosen negative instance. It is calculated by integrating the area under the Receiver Operating Characteristic (ROC) curve, which plots the True Positive Rate (TPR) against the False Positive Rate (FPR) at various threshold settings.

## E. Implementation Details

All experiments were conducted on a system equipped with NVIDIA A100 GPU and AMD EPYC 7473X 24-core processor (48 threads). We train the model for 50 epochs with a learning rate of 0.0005, a weight decay of 0.01, and a batch size of 64. The hidden dimension of all model components is set to 64, and a dropout rate of 0.3 is applied. For each experiment, we run the model with three different random seeds and report the mean and standard deviation of the results. All datasets are split into training, validation, and test sets with a ratio of 6:1:3.

**Architectural Configurations.** To ensure reproducibility, we detail the specific parameterization of AFR-Net's internal modules using the notation defined in the Methodology. The effective resistance encoder $\psi_{\mathrm{res}}(\cdot)$ is implemented as a two-layer MLP with a hidden dimension of 128, utilizing GELU activation to project raw resistance features into the latent space. The edge-gating network $\omega_{\mathrm{gate}}(\cdot)$ employs a two-layer MLP with a hidden dimension of 64, utilizing the SiLU activation function to facilitate gradient flow. For the adaptive flow routing, we set the structural Laplacian regularization scalar $\delta = 10^{-6}$ (consistent with $\mathbf{L}_{\mathrm{flow}} = \mathbf{B}^{\top}\mathbf{C}\mathbf{B} + \delta\mathbf{I}$ in the main text) to guarantee numerical stability, and the mask scaling parameter $\tau = 8.0$. Finally, the prediction head consists of a two-layer MLP with ReLU activation. The framework is implemented using PyTorch[1] and the Deep Graph Library (DGL[2]).

## F. Additional Robustness Analyses

This section provides the robustness analyses referenced from Section 5.

**Cross-validation on the small PPMI cohort.** Because PPMI is small (90 subjects), we further evaluate AFR-Net with 5-fold stratified cross-validation (3 seeds × 5 folds = 15 runs). The cross-validated performance (F1 $82.90 \pm 3.54$, AUC $89.80 \pm 1.56$; Table 5) is consistent with the original held-out split and remains far above the strongest baseline NeuroPath (F1 74.56, AUC 71.88). For the large-scale ABCD cohort (6,381 subjects), the 3-seed protocol is already statistically stable.

---

[1] https://pytorch.org/
[2] https://www.dgl.ai/

*Table 7.* Reliability of the discovered communication patterns. **Left**: Jaccard overlap between AFR-Net's top-100 flow edges and SC-based null models (top-100 edges, 3 checkpoints); near-zero overlap shows the patterns do *not* trivially follow SC weight or degree. **Right**: pairwise Jaccard of the top-100 flow edges across 3 independently initialized models; the high overlap shows the interpretations are robust to initialization.

| Task | Null-model overlap | | Cross-seed consistency | |
|------|---------|-----------|------------|----------|
| | SC weight | SC degree | Avg Jaccard | Pairwise |
| Anx | 0.023 | 0.000 | 0.902 | 0.869/0.966/0.872 |
| OCD | 0.022 | 0.000 | 0.950 | 0.952/0.948/0.949 |

*Table 8.* Performance comparison on ABCD-OCD dataset (mean $\pm$ std %).

| Method | Accuracy | Precision | Recall | F1 | AUC |
|--------|----------|-----------|--------|-----|-----|
| MLP | $61.97 \pm 1.54$ | $\mathbf{63.90 \pm 1.49}$ | $57.47 \pm 3.48$ | $60.48 \pm 2.27$ | $68.62 \pm 0.88$ |
| GCN | $47.50 \pm 2.51$ | $47.24 \pm 3.21$ | $60.34 \pm 20.60$ | $52.26 \pm 10.45$ | $47.11 \pm 3.13$ |
| GAT | $58.23 \pm 2.48$ | $57.38 \pm 2.33$ | $62.64 \pm 1.99$ | $59.90 \pm 2.18$ | $63.58 \pm 1.10$ |
| GIN | $48.07 \pm 4.64$ | $50.19 \pm 7.24$ | $42.82 \pm 21.98$ | $42.93 \pm 11.59$ | $46.97 \pm 6.20$ |
| GraphSAGE | $59.94 \pm 1.94$ | $59.77 \pm 1.92$ | $59.77 \pm 2.49$ | $59.76 \pm 2.12$ | $65.81 \pm 2.12$ |
| Graphormer | $61.95 \pm 2.36$ | $63.15 \pm 1.28$ | $56.32 \pm 7.43$ | $59.40 \pm 4.73$ | $68.01 \pm 1.19$ |
| Triplet | $50.82 \pm 0.67$ | $45.25 \pm 3.45$ | $57.00 \pm 6.18$ | $43.52 \pm 4.14$ | $52.56 \pm 0.03$ |
| AGT | $53.79 \pm 3.44$ | $53.69 \pm 2.66$ | $58.05 \pm 8.13$ | $51.52 \pm 5.47$ | $54.18 \pm 3.17$ |
| BQN | $49.91 \pm 0.12$ | $39.96 \pm 14.20$ | $80.00 \pm 28.28$ | $53.29 \pm 18.91$ | $57.29 \pm 3.26$ |
| BrainGNN | $52.74 \pm 0.61$ | $52.34 \pm 0.53$ | $57.88 \pm 6.15$ | $54.82 \pm 2.79$ | $53.53 \pm 0.10$ |
| Cross-GNN | $56.25 \pm 1.07$ | $55.90 \pm 0.96$ | $62.77 \pm 3.73$ | $58.21 \pm 1.75$ | $58.31 \pm 1.89$ |
| MaskGNN | $58.09 \pm 0.85$ | $58.18 \pm 1.62$ | $61.80 \pm 5.51$ | $58.42 \pm 2.47$ | $63.30 \pm 1.12$ |
| RH-BrainFS | $59.69 \pm 3.12$ | $59.82 \pm 3.13$ | $63.82 \pm 3.65$ | $59.55 \pm 3.15$ | $59.69 \pm 3.12$ |
| NeuroPath | $\underline{62.66 \pm 1.49}$ | $60.60 \pm 1.09$ | $\underline{68.58 \pm 3.61}$ | $\underline{64.13 \pm 2.10}$ | $\underline{69.39 \pm 0.15}$ |
| Ours | $\mathbf{69.10 \pm 4.71}$ | $\underline{63.53 \pm 1.05}$ | $\mathbf{79.60 \pm 5.26}$ | $\mathbf{70.62 \pm 2.36}$ | $\mathbf{72.68 \pm 1.78}$ |

**Edge-capacity ablation (circuit bias vs. learned gate).** To isolate the circuit-theoretic inductive bias from the learned edge gate, we evaluate a *Raw SC Capacity* variant that bypasses the gate and fixes $c_{ij} = \text{SC}_{ij}$. As reported in Table 6, using raw SC as capacity is markedly worse than the full model (e.g., OCD F1 57.31 vs. 70.62) and even underperforms removing routing entirely on OCD and ADHD, while the full model is best on every task. This indicates that the circuit formulation and the adaptive gating are *synergistic*: the Laplacian constraint $\mathbf{L}_{\text{flow}} = \mathbf{B}^\top \mathbf{CB} + \delta\mathbf{I}$ couples all edges globally—so capacities cannot be tuned independently like ordinary per-edge attention weights—while the learned gate adapts these globally constrained capacities to functional context. Neither the inductive bias nor the extra parameters alone explains the improvement.

**Reliability of the discovered patterns.** We assess whether the latent communication patterns are trivial or initialization-dependent. (i) *Null models*: the Jaccard overlap between AFR-Net's top-100 flow edges and SC-based null models (raw SC edge weight; SC degree product) is near zero ($\leq 0.023$; Table 7), so the patterns are not a relabeling of SC magnitude. (ii) *Cross-seed consistency*: the pairwise top-100 Jaccard across 3 independently initialized models is 0.90 (Anx) and 0.95 (OCD), so the interpretations are highly robust to initialization. This robustness is expected, since $\Phi_{ij}$ (Eq. 10) is jointly determined by the data-fixed $\mathbf{L}_{\text{fc}}$ and the structural topology, with only the globally constrained capacities $c_{ij}$ learned.

# G. Full Results of Experiments

Here we report the full experimental results, shown in Tables 8, 9, 10, and 11. As can be seen, our method AFR-Net performs very well across all datasets and achieves overall results that surpass all baselines. This demonstrates that AFR-Net effectively captures the brain's communication patterns, thereby attaining superior performance on downstream tasks.

# H. Additional Interpretability Analysis

To evaluate the generalizability of AFR-Net across different psychiatric conditions, we conduct an identical two-stage analysis on the Obsessive-Compulsive Disorder (OCD) dataset.First, we visualize the informational backbone by extracting the top 100 edges with the highest information flow averaged across the OCD cohort. As shown in Figure 5(a), these

*Table 9.* Performance comparison on ABCD-ADHD dataset (mean ± std %).

| Method | Accuracy | Precision | Recall | F1 | AUC |
|---|---|---|---|---|---|
| MLP | $60.42 \pm 1.44$ | $59.67 \pm 0.75$ | $64.17 \pm 6.29$ | $61.75 \pm 3.17$ | $57.55 \pm 5.06$ |
| GCN | $49.17 \pm 5.64$ | $46.76 \pm 7.73$ | $63.33 \pm 37.86$ | $51.28 \pm 21.88$ | $50.81 \pm 9.23$ |
| GAT | $53.75 \pm 2.17$ | $52.88 \pm 1.59$ | $68.33 \pm 5.20$ | $59.59 \pm 2.77$ | $59.31 \pm 1.37$ |
| GIN | $47.92 \pm 1.91$ | $48.46 \pm 1.43$ | $\mathbf{74.17 \pm 16.07}$ | $58.31 \pm 5.85$ | $44.37 \pm 4.37$ |
| GraphSAGE | $47.92 \pm 1.91$ | $47.72 \pm 2.07$ | $48.33 \pm 10.10$ | $47.81 \pm 5.84$ | $52.60 \pm 3.32$ |
| Graphormer | $55.28 \pm 1.90$ | $59.72 \pm 2.48$ | $59.97 \pm 6.96$ | $55.19 \pm 5.09$ | $58.79 \pm 2.17$ |
| Triplet | $49.63 \pm 2.17$ | $53.15 \pm 6.77$ | $65.75 \pm 3.72$ | $47.91 \pm 5.07$ | $48.84 \pm 3.72$ |
| AGT | $55.67 \pm 2.44$ | $56.75 \pm 4.11$ | $41.29 \pm 12.78$ | $41.91 \pm 7.66$ | $53.55 \pm 4.10$ |
| BQN | $50.17 \pm 0.24$ | $40.08 \pm 14.02$ | $80.00 \pm 28.28$ | $53.41 \pm 18.75$ | $57.11 \pm 0.68$ |
| BrainGNN | $55.00 \pm 2.57$ | $55.88 \pm 3.58$ | $47.50 \pm 19.47$ | $51.35 \pm 20.84$ | $54.37 \pm 5.89$ |
| Cross-GNN | $54.31 \pm 2.97$ | $54.90 \pm 2.52$ | $53.72 \pm 5.82$ | $52.76 \pm 3.79$ | $56.24 \pm 3.94$ |
| MaskGNN | $56.58 \pm 2.20$ | $\underline{61.51 \pm 4.22}$ | $45.48 \pm 1.91$ | $49.22 \pm 1.50$ | $61.58 \pm 2.05$ |
| RH-BrainFS | $\underline{62.48 \pm 3.63}$ | $\mathbf{62.31 \pm 2.82}$ | $69.84 \pm 9.22$ | $61.59 \pm 2.94$ | $61.76 \pm 2.61$ |
| NeuroPath | $58.33 \pm 3.82$ | $56.92 \pm 3.47$ | $69.17 \pm 6.29$ | $\underline{62.36 \pm 3.71}$ | $\mathbf{64.19 \pm 0.35}$ |
| Ours | $\mathbf{63.75 \pm 2.60}$ | $60.86 \pm 2.61$ | $\underline{71.67 \pm 9.46}$ | $\mathbf{65.50 \pm 2.35}$ | $\underline{62.38 \pm 1.08}$ |

*Table 10.* Performance comparison on ABCD-Anx dataset (mean ± std %).

| Method | Accuracy | Precision | Recall | F1 | AUC |
|---|---|---|---|---|---|
| MLP | $53.14 \pm 1.11$ | $52.89 \pm 0.98$ | $53.22 \pm 3.88$ | $53.03 \pm 2.44$ | $56.83 \pm 0.65$ |
| GCN | $45.47 \pm 2.15$ | $46.07 \pm 2.78$ | $62.75 \pm 20.17$ | $52.47 \pm 8.95$ | $45.02 \pm 3.33$ |
| GAT | $54.11 \pm 2.66$ | $53.69 \pm 2.48$ | $56.30 \pm 4.68$ | $54.95 \pm 3.45$ | $54.37 \pm 1.51$ |
| GIN | $48.81 \pm 1.21$ | $48.33 \pm 1.44$ | $53.50 \pm 23.53$ | $49.50 \pm 10.60$ | $48.01 \pm 0.78$ |
| GraphSAGE | $54.39 \pm 0.72$ | $54.29 \pm 0.78$ | $53.22 \pm 0.49$ | $53.75 \pm 0.54$ | $56.15 \pm 1.84$ |
| Graphormer | $50.09 \pm 0.86$ | $50.08 \pm 0.69$ | $54.08 \pm 1.40$ | $51.34 \pm 0.87$ | $51.71 \pm 0.27$ |
| Triplet | $49.50 \pm 0.55$ | $36.68 \pm 5.12$ | $53.81 \pm 13.17$ | $39.56 \pm 7.83$ | $48.07 \pm 0.91$ |
| AGT | $52.93 \pm 1.75$ | $54.50 \pm 2.83$ | $36.41 \pm 12.74$ | $38.54 \pm 9.82$ | $52.51 \pm 1.10$ |
| BQN | $50.03 \pm 0.04$ | $40.01 \pm 14.12$ | $\mathbf{80.00 \pm 28.28}$ | $53.35 \pm 18.84$ | $51.30 \pm 0.69$ |
| BrainGNN | $53.70 \pm 2.06$ | $54.53 \pm 2.55$ | $42.58 \pm 4.14$ | $47.71 \pm 3.10$ | $55.38 \pm 4.06$ |
| Cross-GNN | $49.66 \pm 0.36$ | $49.49 \pm 0.55$ | $52.79 \pm 3.01$ | $50.08 \pm 2.05$ | $50.56 \pm 1.25$ |
| MaskGNN | $51.09 \pm 1.75$ | $51.53 \pm 1.30$ | $50.80 \pm 5.64$ | $50.36 \pm 3.40$ | $52.67 \pm 0.72$ |
| RH-BrainFS | $54.69 \pm 5.41$ | $\underline{55.33 \pm 6.12}$ | $69.50 \pm 8.11$ | $53.69 \pm 5.26$ | $54.75 \pm 5.36$ |
| NeuroPath | $\underline{56.49 \pm 2.33}$ | $\mathbf{56.54 \pm 2.44}$ | $54.62 \pm 2.22$ | $\underline{55.56 \pm 2.24}$ | $\underline{57.65 \pm 2.07}$ |
| Ours | $\mathbf{58.16 \pm 1.26}$ | $55.07 \pm 2.35$ | $\underline{72.27 \pm 10.53}$ | $\mathbf{62.15 \pm 2.41}$ | $\mathbf{59.03 \pm 1.94}$ |

*Table 11.* Performance comparison on PPMI dataset (mean ± std %).

| Method | Accuracy | Precision | Recall | F1 | AUC |
|---|---|---|---|---|---|
| MLP | $56.48 \pm 5.78$ | $64.58 \pm 5.51$ | $46.67 \pm 10.41$ | $54.00 \pm 9.10$ | $70.52 \pm 0.79$ |
| GCN | $55.56 \pm 2.78$ | $56.56 \pm 2.43$ | $\underline{88.33 \pm 7.64}$ | $68.79 \pm 1.37$ | $64.11 \pm 5.57$ |
| GAT | $62.96 \pm 2.62$ | $63.75 \pm 3.13$ | $78.33 \pm 2.36$ | $70.19 \pm 0.88$ | $70.83 \pm 4.25$ |
| GIN | $55.56 \pm 0.00$ | $55.56 \pm 0.00$ | $\mathbf{100.00 \pm 0.00}$ | $71.43 \pm 0.00$ | $50.00 \pm 0.00$ |
| GraphSAGE | $60.19 \pm 8.02$ | $58.54 \pm 5.17$ | $\mathbf{100.00 \pm 0.00}$ | $73.76 \pm 4.04$ | $70.31 \pm 7.15$ |
| Graphormer | $64.81 \pm 3.21$ | $64.26 \pm 3.13$ | $\underline{83.33 \pm 2.89}$ | $72.49 \pm 1.55$ | $61.77 \pm 5.02$ |
| Triplet | $51.85 \pm 5.24$ | $53.70 \pm 2.62$ | $70.00 \pm 42.43$ | $53.17 \pm 25.82$ | $66.25 \pm 9.51$ |
| AGT | $53.70 \pm 2.62$ | $55.79 \pm 0.33$ | $81.67 \pm 25.93$ | $64.29 \pm 10.10$ | $61.20 \pm 1.69$ |
| BQN | $52.78 \pm 2.27$ | $59.07 \pm 5.43$ | $68.33 \pm 34.24$ | $56.36 \pm 18.10$ | $60.73 \pm 13.60$ |
| BrainGNN | $67.90 \pm 4.62$ | $71.03 \pm 3.41$ | $71.11 \pm 6.29$ | $71.02 \pm 4.62$ | $71.30 \pm 4.41$ |
| Cross-GNN | $62.96 \pm 1.31$ | $70.78 \pm 5.30$ | $60.00 \pm 14.14$ | $63.51 \pm 5.42$ | $65.83 \pm 8.74$ |
| MaskGNN | $68.52 \pm 3.46$ | $69.80 \pm 5.05$ | $78.33 \pm 4.71$ | $73.50 \pm 1.11$ | $68.23 \pm 0.74$ |
| RH-BrainFS | $70.37 \pm 3.20$ | $69.83 \pm 3.72$ | $80.95 \pm 4.76$ | $68.54 \pm 3.15$ | $68.25 \pm 3.06$ |
| NeuroPath | $\underline{72.22 \pm 0.00}$ | $\underline{75.93 \pm 1.60}$ | $73.33 \pm 2.89$ | $\underline{74.56 \pm 0.76}$ | $\underline{71.88 \pm 1.13}$ |
| Ours | $\mathbf{82.69 \pm 3.33}$ | $\mathbf{84.37 \pm 0.43}$ | $\underline{83.33 \pm 8.25}$ | $\mathbf{83.70 \pm 3.85}$ | $\mathbf{90.38 \pm 1.64}$ |

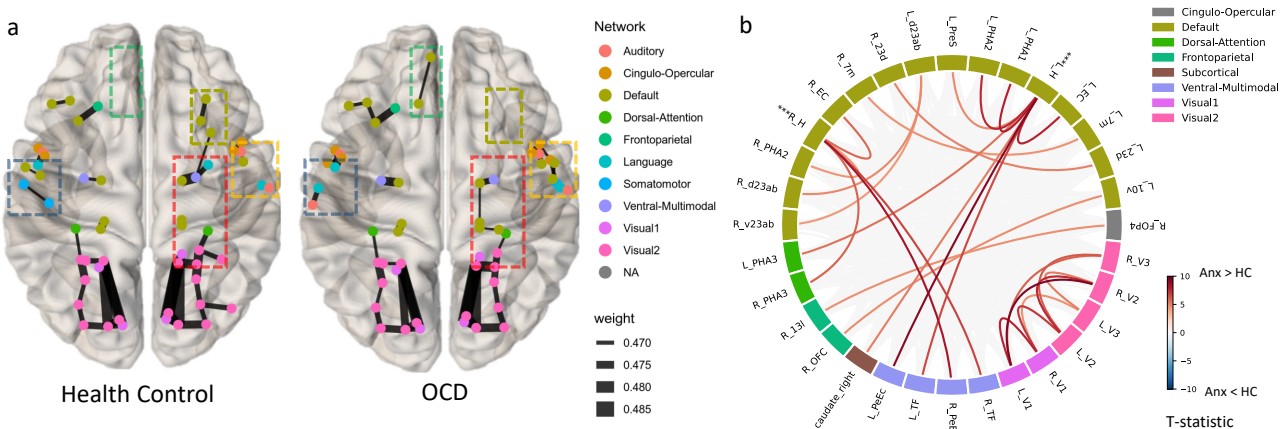

*Figure 5.* (a) The Informational Backbone. Visualization of the top 100 edges with the highest information flow intensity averaged across all subjects in the OCD dataset. Consistent with the main findings, the model identifies the visual and somatomotor networks as the stable structural core. (b) Pathological Hyper-connectivity in OCD. A circle plot showing edges with significant group differences ($p < 0.05$, FDR corrected). Red lines indicate hyper-active flow (Patient > HC). The Primary Visual Cortex (V1/V2) and Hippocampus (H, marked with ***) emerge as pathological hubs, exhibiting dense hyper-connectivity between sensory processing and memory encoding circuits.

high-traffic edges remain predominantly concentrated within the visual and somatomotor networks. This observation mirrors the topological findings in the Anxiety dataset and aligns with the brain's conserved structural core (Bullmore & Sporns, 2009), confirming that AFR-Net consistently learns valid anatomical priors regardless of the specific disease pathology.Next, we investigate pathological routing alterations using a two-sample $t$-test ($p < 0.05$, FDR). Figure 5(b) reveals a distinct pattern of Visual-Limbic Hyper-connectivity (Patient > HC). Specifically, the Primary Visual Cortex (V1, V2) and the Hippocampus (H) emerge as coupled pathological hubs, exchanging excessive information. Synthesizing our connectivity findings with this functional evidence offers a mechanistic explanation for the "doubting disease." Existing neuroscience literature posits that OCD is driven by a deficit in metamemory, leading to the "Memory Distrust" phenomenon (van den Hout & Kindt, 2003). While the visual cortex is responsible for gathering sensory evidence, the hippocampus is critical for generating the "feeling of knowing" that terminates the action. The hyper-active flow we observed from Visual to Limbic regions strongly suggests a maladaptive sensory-memory mismatch: the relentless accumulation of visual evidence fails to translate into memory confidence, trapping the patient in a loop of repetitive checking.This finding is highly consistent with recent high-precision neuroimaging studies. For instance, Hearne et al. (2025) identify visual network hyper-connectivity as a primary discriminative feature of OCD, while Liu et al. (2025) report specific aberrant coupling between occipital and subcortical regions. By autonomously uncovering this specific "Check-and-Doubt" short-circuit, AFR-Net demonstrates its sensitivity in disentangling disease-specific biomarkers from the shared brain topology.

