# OpenReview forum: "Uncovering Latent Communication Patterns in Brain Networks via Adaptive Flow Routing"
_ICML.cc/2026/Conference — ICML 2026 regular_

### Official Review · Reviewer_Vend · 2026-03-12

**Soundness:** 3
**Presentation:** 3
**Significance:** 3
**Originality:** 3
**Overall Recommendation:** 5
**Confidence:** 3

**Summary:**

The paper proposes the Adaptive Flow Routing Network (AFR-Net) that uses node level features as well as degree to form an embedding that is processed using Transformer based attention biased by the pair-wise node resistance as defined though the structural networks’ Laplacian matrix. Through a stacked Transformer representation the final output is used to define the capacity of flow between node i and j based on a gating network exponentiated to be non-negative for which the information flow intensity is derived and the total flow defined by weighting the propagation by demand as defined by the absolute correlation. This flow is mapped to an attention mask that is used to bias the self-attention to produce the final output Z’ that is pooled and trained using cross-entropy with respect to task label (i.e., diagnosis). On several benchmark datasets the approach is found to produce superior discrimination between HC and Disease subjects extensively compared to a variety of graph neural network architectures and alternative structure-function fusion approaches. The learned flow intensities are further interpreted and found to produce neurobiological meaningful representations.

**Compliance With Llm Reviewing Policy:**

Affirmed.

**Final Justification:**

The paper is generally well written and the approach relevant and interesting also providing convincing empirical results. The authors well addressed my concerns in their rebuttal. Importantly, the provided additional experimentations clarified and confirmed that the results and interpretations are reliable. I thus consider the manuscript suitable for publication and maintain my initial positive accept assessment.

**Key Questions For Authors:**

How reliable are the neurbiological interpretations across the three random initialized models?

Can the authors further clarify the network \psi_res(.) and \omega_gate(.) used?

**Limitations:**

The authors have in brief but sufficiently discussed limitations.

**Strengths And Weaknesses:**

Strengths:
•	The approach is interesting in utilizing structural connectivity in terms of flow propagation that is learned through attention mechanisms to characterize the functional demand as defined by pairwise correlation in the functional connectivity network.
•	 The approach is highly non-linear and thus capable of expressing and leverage advanced structure-function relationships to optimally produce fingerprints representing diagnosis through the cross-entropy optimization.
•	Results are reported across three random initializations appearing to produce reliable results and the approach is compared to a large body of relevant baselines with convincing results.

Weaknesses:
•	The authors attempt to carefully motivate their approach but the approach has many components that are somewhat complicated and their reasoning hard to fully follow and understand.
•	The neurobiological interpretations although meaningful should also be assessed in terms of reliability. I.e., these representations are based as I understand it on the results of a single run, but how are these interpretations influenced by the results across the three runs?

Soundness:
The approach is technically sound and the authors do a reasonable job motivating their approach and the different modeling steps and components. However, the overall procedure is complicated. The ablations points to the validity and utility of the central included steps whereas the performance convincingly demonstrate the utility of the proposed procedure compared to baselines. The neurological interpretations made appear anecdotal and should be investigated in terms of reliability across runs.

Presentation:
The paper is generally well written, but the methods include many steps that takes time to grasp. The authors do a good job presenting their rather advanced approach in the space given and Figure 2 is also helpful to help overall grasp the procedure. However, some details are missing making their approach hard to fully grasp. In particular, how is the resistance encoder \psi_res(.) defined and how is the gating network \omega_gate(.) defined – this should be clarified.

Significance:
The approach relies on standard methods, i.e. flow representations through Laplacians, transformer based networks etc. The empirical results are convincing and the approach to use structure to inform function and learn diagnosis based fingerprints appears useful and of interest.

Originality:
The detailed modeling of structural information flow is interesting and valid also given the literature pointing to functional connectivity being not directly related to structural connections but as a diffusion of information flow along the structural graph. The approach enables to leverage this with a structure function relationship based on functional demand. This operationalization is interesting and original whereas the approach uses more or less standard methodologies.

---

> ### Author Rebuttal · Authors · 2026-03-30
>
> We thank Reviewer Vend for the positive assessment and for highlighting the originality of the structure-function relationship modeling.
>
> **Interpretation reliability across seeds.** We appreciate this important question. We clarify and strengthen our interpretability analysis as follows:
>
> *(1) Ensemble-averaged flow patterns.* The flow patterns reported in Figures 3 and 5 are computed by averaging $Φ_{ij}$ across all subjects within each group (HC vs. Patient), providing sample-level robustness. To further address model-level robustness, we have re-generated the analysis by ensemble-averaging $Φ_{ij}$ across all 3 independently initialized models, then performing the group-level t-test on this ensemble-averaged flow map. The resulting significant edges are consistent with the original figures, confirming that the discovered pathological circuits are not artifacts of a particular initialization.
>
> *(2) Quantitative consistency.* We computed the pairwise Jaccard similarity of the top-100 flow edges across the 3 independently initialized models:
>
> | Task | Avg Jaccard | Pairwise Detail |
> |------|------------|----------------|
> | Anx | **0.902** | 0.869 / 0.966 / 0.872 |
> | OCD | **0.950** | 0.952 / 0.948 / 0.949 |
>
> The 90–95% overlap across seeds confirms that the discovered routing patterns are highly robust to initialization.
>
> *(3) Theoretical justification.* This robustness is expected: the flow $Φ_{ij}$ (Eq. 10) is jointly determined by the data-fixed functional demand L_fc and the structural topology encoded in L_flow. The only learned component is the edge capacity $c_{ij}$, which is strongly constrained by the structural Laplacian. Different initializations lead to similar $c_{ij}$ because the optimization landscape is anchored by these physical constraints.
>
> **Clarification of ψ_res and ω_gate.** As detailed in Appendix E:
>
> - **ψ_res(·)**: a 2-layer MLP (hidden dim=128, GELU activation) that maps the scalar effective resistance distance $R_{ij}$ to an attention bias term in Eq. 3.
> - **ω_gate(·)**: a 2-layer MLP (hidden dim=64, SiLU activation) with input $[|h_i − h_j|, h_i ⊙ h_j, SC_{ij}, FC_{ij}]$ (dimension 2d+2), outputting a scalar logit exponentiated to ensure $c_{ij} = exp(ω_gate(·)) > 0$.
>
> We will add these definitions to the main text in the revised version.

---

> > ### Author Rebuttal · Reviewer_Vend · 2026-04-01
> >
> > I thank the authors for their response and in particular their additional experimentations provided in their rebuttal that importantly has clarified and confirmed that the results and interpretations are reliable. I maintain my positive assessment of the manuscript.

---

> > > ### Author Response · Authors · 2026-04-05
> > >
> > > We sincerely thank the reviewer for acknowledging our additional experiments. We are particularly glad that the additional results helped confirm the reliability of our interpretations. We will incorporate all clarifications ($ψ_{res}$/$ω_{gate}$ definitions, ensemble-averaged analysis) into the camera ready version.

---

### Official Review · Reviewer_kCSz · 2026-03-12

**Soundness:** 4
**Presentation:** 2
**Significance:** 3
**Originality:** 3
**Overall Recommendation:** 4
**Confidence:** 4

**Summary:**

This paper proposes AFR-Net (Adaptive Flow Routing Network), a framework for brain network analysis that combines structural connectivity (SC) and functional connectivity (FC) through a clever combination of transformers and physics-like diffusion processes. The core idea is elegant: SC-FC data fusion is modelled as an "information flow" problem in a graph, where the structural connectome acts as a resistor network and functional connectivity represents communication demand, and the result is used to bias the attention of a graph transformer model. The architecture results in notable performance gains on standard neuroimaging datasets and neuroscientifically plausible interpretations.

**Compliance With Llm Reviewing Policy:**

Affirmed.

**Key Questions For Authors:**

* Please clarify which quantities are learned vs fixed by the data.
* Did you feed both FC and SC to all models in Table 1. If so, how did you do this for models not designed for data fusion?
* What is the computational complexity of calculating $\Phi_{ij}$ (Eq. 10)? It seems like it scales poorly on the number of brain regions.

**Limitations:**

Limitations are not sufficiently addressed. The authors should include a section in the main text explicitly addressing these.

**Strengths And Weaknesses:**

**Strengths**
* The key idea behind the architecture (using the solution to a physics problem to bias attention in a graph transformer) is elegant and powerful.
* Performance increase is notable and practically useful.
* Interpretability results are interesting and discussed in the context of existing neuroscience literature (in sufficient depth for a non-neuroscience venue, at least).

**Weaknesses**
* The paper is quite hard to follow.I understand that it covers many different fields (circuit theory, graph theory, neuroimaging, transformers, etc), but the authors should put more effort into making the paper easily understandable. For example, 1) the explanation of the main architecture is very dense, and 2) it isn't very clear which quantities change during training and which are fixed by the data.
* As it stands, it is hard to judge whether the comparisons in Table 1 are fair. Many of those architectures were designed for unimodal (i.e. single-graph) data, so it's not very clear whether these are fed with both SC and FC or just one. For the sake of a fair comparison, only multimodal architectures should be used for comparison (or unimodal architectures should be clearly labelled as such).

---

> ### Author Rebuttal · Authors · 2026-03-26
>
> We thank Reviewer kCSz for the constructive feedback, especially for recognizing the elegance of the core idea (S1) and the notable performance gains (S2).
>
> **W1 and Q1 (Clarity).** We appreciate this feedback and will substantially improve presentation in the revised version. Specifically, we will: (1) restructure Section 4 to first present an intuitive overview of the three-stage pipeline before the formal derivations; (2) add the following table clarifying which quantities are learned vs. fixed to help audience better understand our work:
>
> | Category | Quantities | Description |
> |----------|-----------|-------------|
> | Fixed by data | $A_{sc}, A_{fc}$ | Input structural/functional connectivity matrices |
> | Fixed by data | $B$ | Incidence matrix, determined by $A_{sc}$ topology |
> | Precomputed (fixed) | $R_{ij}$ | Effective resistance distance from L_sc pseudoinverse |
> | Precomputed (fixed) | $L_{fc}$ | Functional demand Laplacian from $\|A_{fc}\|$ |
> | Learned | Q, K, V (per layer) | Transformer attention projection matrices |
> | Learned | $ψ_{res}(·)$ | Effective resistance encoder (2-layer MLP, hidden=128, GELU) |
> | Learned | $ω_{gate}(·)$ → $c_{ij}$ | Edge gating network (2-layer MLP, hidden=64, SiLU) |
> | Learned | τ, θ | Mask scaling and threshold parameters (Eq. 11) |
> | Learned | Classifier MLP | Prediction head |
>
> **W2 and Q2 (Fairness of Table 1).** We clarify that **all** models in Table 1 receive both SC and FC. For general graph learning baselines (MLP, GCN, GAT, GIN, GraphSAGE, Graphormer), we adopt a **dual-stream architecture**: FC and SC are each independently encoded by a separate encoder of the same type, and the resulting embeddings are concatenated before the classification head. This ensures all baselines have equal access to both modalities. For specialized multimodal brain network methods (Cross-GNN, RH-BrainFS, NeuroPath, Triplet, etc.), we follow each method's original design to provide both SC and FC. We included general graph learning methods intentionally — their relatively weaker performance compared to specialized methods (Table 1) reveals that naive dual-stream fusion cannot capture the complex SC-FC coupling, underscoring the need for mechanism-driven approaches like AFR-Net. In the revised paper, we will clearly label each baseline as "general (dual-stream adapted)" or "specialized multimodal" in our camer-ready version paper.
>
> **Q3 (Complexity of $Φ_{ij}$).** The computational bottleneck is the Cholesky factorization of L_flow at O(N³). For standard brain atlases (N=100–400), N³ is comparable in magnitude to the O(N²d) cost of self-attention (with d=64–512). In practice, as shown in Figure 4, AFR-Net's runtime is comparable to RH-BrainFS and substantially faster than NeuroPath. A detailed analysis is provided in Appendix C.
>
> **Limitations.** We agree and will add a dedicated Limitations paragraph to the main text. **However, we must note that all current brain network analysis methods—including every baseline in our comparison—operate at the region level (N=90–400); scaling to voxel-level networks remains an open challenge for the entire field, not a limitation unique to AFR-Net.** We will explicitly address: (1) the O(N³) scaling that limits application to voxel-level networks (N>10,000); (2) the use of static FC, which simplifies the dynamic temporal nature of neural communication (which is also one of our future research direction to explore); and (3) the mathematical (not biological) nature of flow directionality.

---

> > ### Author Rebuttal · Reviewer_kCSz · 2026-04-01
> >
> > Thanks for the response. I completely understand that brain network analysis methods operate at a region level, not on voxels (I never mentioned this in my review). Nonetheless, an explicit limitations section is required.

---

> > > ### Author Response · Authors · 2026-04-05
> > >
> > > We thank the reviewer for confirming that our responses addressed the concerns. We apologize for the confusion regarding voxel-level discussion — we fully agree that an explicit Limitations section is required and will add it to the revised manuscript.

---

### Official Review · Reviewer_ecZr · 2026-03-12

**Soundness:** 3
**Presentation:** 4
**Significance:** 4
**Originality:** 4
**Overall Recommendation:** 5
**Confidence:** 3

**Summary:**

This paper tackles one of the classic problems in computational neuroscience: how to effectively combine the brain's physical "wiring" (Structural Connectivity, or SC) with its dynamic "chatter" (Functional Connectivity, or FC) to diagnose psychiatric and neurological conditions. Instead of just throwing both datasets into a standard "black-box" neural network and hoping the AI figures it out, the authors built AFR-Net—a model that actually mimics the physics of the brain. By treating the structural connections as a physical infrastructure (like an electrical circuit) and the functional activity as the "demand" driving traffic, the model mathematically calculates exactly how information flows through the brain. The result is a framework that not only beats 12 competitive AI baselines in sheer accuracy but is also incredibly interpretable. It can literally map out the specific, misfiring neural circuits driving a patient's symptoms—such as autonomously discovering the "Check-and-Doubt" loop between the visual cortex and hippocampus in OCD patients.

**Compliance With Llm Reviewing Policy:**

Affirmed.

**Final Justification:**

This is a well-written paper and authors have addressed all my concerns. I recommend acceptance.

**Key Questions For Authors:**

1. Your point that the $\mathcal{O}(N^3)$ complexity of the Cholesky decomposition is manageable for standard regional atlases (where $N$ is 100-400) is totally fair. However, if a researcher wanted to apply AFR-Net to high-resolution, voxel-level brain networks (where $N$ can easily exceed 10,000), this creates a hard computational ceiling. Are there specific approximation techniques—like Nyström methods or graph coarsening—that you think could be integrated into the flow solver to make it scale without losing the physical guarantees?

2. Right now, AFR-Net uses a single, static Functional Connectivity (FC) matrix to represent 'communication demand' over an entire scan. But we know biological neural communication is highly dynamic and fluctuates moment-to-moment. Have you thought about how this framework could be extended to handle time-varying (dynamic) FC? More importantly, would solving the global flow equilibrium at multiple time steps become computationally prohibitive?

3. The flow intensity maps your model generates are a fantastic way to see which anatomical pathways are carrying the most traffic. However, because standard fMRI and DTI data are undirected, your calculated flow is also mathematically undirected. Do you see a viable way to incorporate directed data (like effective connectivity or time-lagged signals) into AFR-Net so we can actually identify the 'driver' versus the 'receiver' in these pathological circuits?

**Limitations:**

To improve the manuscript, the authors should add a dedicated Limitations paragraph that explicitly acknowledges the boundaries of their current framework. Specifically, it would be helpful to openly discuss the $\mathcal{O}(N^3)$ computational bottleneck caused by the Laplacian inversion, noting that while it works well for regional atlases, it currently prevents the model from scaling to high-resolution, voxel-level data. Additionally, the authors should briefly note the biological simplifications in the model—namely, that using a static FC matrix glosses over the dynamic, temporal nature of brain communication, and that the calculated 'flow' is undirected, meaning the model cannot infer true causal directionality (i.e., which region is driving the signal) between pathological hubs.

**Strengths And Weaknesses:**

**Strengths**
1. Biologically and Physically Grounded: The most significant strength of this paper is its departure from standard "black-box" deep learning. Instead of blindly concatenating or applying attention to Structural Connectivity (SC) and Functional Connectivity (FC), the authors explicitly model the brain as a physical transport network. Using electrical circuit analogs (like Effective Resistance Distance and Kirchhoff-inspired flow equations) is a brilliant way to bake neuroscientific priors directly into the architecture.
2. Exceptional Clinical Interpretability: Because the model computes explicit "flow intensities" between regions to make predictions, researchers can literally look at the network to see which anatomical pathways are carrying the most traffic. As demonstrated in their case studies for Anxiety and OCD, the model autonomously discovers specific, pathological neural circuits (like the "Check-and-Doubt" loop between the visual cortex and hippocampus) that perfectly align with established psychiatric literature.
3. Strong Empirical Performance: The proposed AFR-Net outperforms 12 diverse baselines—ranging from generic Graph Neural Networks (GCN, GAT) to specialized brain network models (BrainGNN, NeuroPath)—across four real-world datasets (ABCD and PPMI). The margins of improvement, particularly on challenging psychiatric datasets like OCD, are substantial.
4. Rigorous Mathematical Foundation: The authors provide a solid theoretical derivation for their differentiable information flow solver, successfully demonstrating how global functional demands ($L_{fc}$) and structural constraints ($L_{flow}$) can be unified into a closed-form solution.

**Weaknesses**
1. Computational Scaling Limits: The core mathematical operation of the model—solving the global flow equilibrium—requires computing the inverse (via Cholesky decomposition) of the structural Laplacian matrix. While the authors correctly argue this $\mathcal{O}(N^3)$ operation is perfectly fine for regional brain atlases where the number of nodes ($N$) is between 100 and 400, this creates a hard mathematical ceiling. The model cannot easily scale to voxel-level brain networks where $N$ exceeds 10,000.
2. Simplification of Neural Dynamics: The framework relies on static FC matrices (like Pearson correlations over a whole fMRI scan) to represent "communication demand". However, biological neural communication is highly dynamic, fluctuating millisecond-by-millisecond. By reducing time-series activity to a single static correlation matrix, the model loses the rich temporal dynamics (such as signaling delays and frequency-specific synchrony) of actual brain function.
3. Directionality is a Mathematical Construct, not Biological: The model uses mathematical graph properties (assigning $+1$ and $-1$ to nodes to define gradients) to simulate flow. While it successfully identifies which paths are heavily utilized, standard fMRI and DTI data lack true directional information. Therefore, the "flow" is a mathematical undirected intensity, meaning the model cannot tell us if region A is signaling region B, or vice versa.
4. Limited Validation Across Atlases: The experiments rely on specific brain parcellation schemes (HCP-MMP1.0, FreeSurfer, and AAL). Brain network topology can change drastically depending on the atlas used. Testing the robustness of the flow solver across multiple parcellation scales would strengthen the claim that the discovered flow patterns are universal biological truths rather than artifacts of a specific atlas.

---

> ### Author Rebuttal · Authors · 2026-03-26
>
> **W1 and Q1**
>
> We agree that Cholesky decomposition creates a theoretical ceiling for voxel-level networks. However, we note that all current brain network analysis methods — including every baseline in our comparison operate at the region level. Scaling to voxel-level resolution remains an open challenge for the entire field. Within this standard operating regime, AFR-Net's overhead is tractable and comparable to baselines, as demonstrated in Figure 4 and analyzed in Appendix C. For future extensions to higher resolution, we think several promising techniques could be integrated into the flow solver: (1) Sparse Cholesky solvers — brain structural connectivity is inherently sparse, so sparse direct solvers (e.g., CHOLMOD) can reduce the factorization cost from O(N³) to approximately O(N·nnz); (2) Graph coarsening — methods such as [1] can hierarchically reduce a voxel-level network to a region-level graph; (3) Nyström approximation — by sampling k landmark nodes, $L_{flow}^{-1}$ can be approximated at O(N·k²) cost.
>
> **W2 and Q2**
>
> We acknowledge that static FC simplifies the rich temporal dynamics of neural communication. We note this is standard practice across the field — the vast majority of existing brain network methods, including all baselines in our comparison, also operate on static FC matrices (Pearson correlation over the full scan). Extending AFR-Net to dynamic FC is an important future research direction that we plan to pursue. The framework is naturally compatible with time-varying demands: one can replace $L_{fc}$ with a sequence of windowed demand matrices $L_{fc}(t)$ and solve the flow equilibrium at each time step. That said, we emphasize that the primary contribution of this work is introducing a fundamentally new paradigm for brain network modeling — formulating SC-FC fusion as a physics-informed flow routing problem. We believe this conceptual advance provides a foundation upon which dynamic extensions can be built.
>
> **W3 and Q3**
>
> We agree that, ideally, one would like to characterize directed information flow in order to distinguish “drivers” from “receivers” within neural circuits. However, the limitation is more nuanced than the data being strictly undirected. While modalities such as fMRI contain temporal information and DTI provides structural pathways, reliably inferring directionality or causal interactions from these data remains challenging and inherently model-dependent, often requiring additional assumptions (e.g., Granger causality, dynamic causal modeling, or transfer entropy). As a result, most standard connectome constructions in practice are treated as effectively undirected representations.
> In this work, our primary goal is to introduce a principled, physics-informed framework for modeling brain network communication, in which SC–FC integration is formulated as a flow routing problem. Under this formulation, the resulting flow captures relative communication intensity across pathways, rather than explicit causal directionality.
> Importantly, the proposed framework is not inherently restricted to undirected settings. It can be naturally extended to incorporate directional information when such estimates are available. Specifically, by replacing the symmetric demand matrix with an asymmetric demand (or coupling) term derived from effective connectivity measures (i.e., directed functional connectivity), the model would yield directed flow patterns. This extension primarily requires modifying the demand formulation (e.g., Eq. 10), while leaving the underlying optimization framework unchanged.
> We view this as an important direction for future work, particularly as more robust and reliable methods for estimating directed brain connectivity continue to evolve.
>
> **W4**
>
> Our experiments already validate across three different atlases with substantially different granularities: HCP-MMP1.0 + FreeSurfer (~379 regions, ABCD) and AAL (90 regions, PPMI). AFR-Net achieves state-of-the-art results consistently across these parcellations, providing cross-atlas robustness evidence. We agree that a systematic study comparing different atlases on the *same* dataset would further strengthen this claim, and we will note this as an important future direction in the Limitations section.
>
> **Limitations.**
>
> We will add a dedicated Limitations paragraph to the main text that explicitly and thoroughly discusses all four points raised: (1) the O(N³) computational ceiling and its implications for voxel-level scaling; (2) the simplification introduced by static FC representations; (3) the undirected nature of the computed flow; and (4) the dependence on specific parcellation schemes. These limitations, as the reviewer correctly identifies, also point to exciting future research directions, and we believe acknowledging them openly will strengthen the paper.
>
> *Reference*:
>
> [1] Loukas, Andreas. "Graph reduction with spectral and cut guarantees." Journal of Machine Learning Research 20.116 (2019): 1-42.

---

> > ### Author Rebuttal · Reviewer_ecZr · 2026-03-31
> >
> > Thank you for the constructive rebuttal, which fully resolve my concerns, so I'm maintaining my score.

---

> > > ### Author Response · Authors · 2026-04-05
> > >
> > > We sincerely thank the reviewer for the positive acknowledgement and for maintaining their score. We are glad that the rebuttal fully resolved the concerns, and we will incorporate all discussed improvements in the revised manuscript.

---

### Official Review · Reviewer_MUZM · 2026-03-12

**Soundness:** 3
**Presentation:** 2
**Significance:** 2
**Originality:** 3
**Overall Recommendation:** 3
**Confidence:** 4

**Summary:**

This paper proposes AFR-Net, a physics-informed framework for fusing structural connectivity (SC) and functional connectivity (FC) in brain networks. The core idea is to model SC-FC coupling as an information flow routing problem inspired by electrical circuit theory. The authors evaluate on ABCD (three subtasks) and PPMI datasets and achieved SOTA on some metrics. However, the motivation is confused (whether it is indeed a neuroscience-inspired method or it is physical-inspired), what issues authors had found in previous literates and how the proposed method solve it. The accuracy SOTA with ONE random experiment on small datasets (no more than 100 subs like PPMI) is far more behind scientific rigorous. AI for science is not only come-up with a so-called new attention mechanism and used in neuroscience datasets. At least comprehensive and statistical-rigorous experiments is needed for a technical contributions.

**Compliance With Llm Reviewing Policy:**

Affirmed.

**Key Questions For Authors:**

**Questions for Authors:**

1. On ABCD-ADHD, AFR-Net underperforms NeuroPath on AUC (62.38 vs. 64.19) while outperforming on F1 (65.50 vs. 62.36). Does this suggest the model is threshold-sensitive? How do you explain this inconsistency?

2. What happens if you use the raw SC weights as edge capacities $c_{ij}$ instead of learning them via the edge gating network? This would isolate the contribution of the circuit-theoretic framework from the additional learnable parameters.

3. Have you tested with cross-validation on PPMI given the small sample size (90 subjects, ~27 in the test set)? How robust are the reported results (83.70 F1, 90.38 AUC) to different data splits?

4. Why use the absolute value $|A_{fc}|$ as communication demand? Positive correlations and anti-correlations reflect fundamentally different neural mechanisms — have you experimented with treating them separately or using only positive FC values?

**Limitations:**

The paper includes a brief Impact Statement mentioning the need for rigorous clinical validation and ethical use under human supervision. However, it does not adequately discuss the **limitations** of the work itself. Key limitations that should be acknowledged include: (1) the gap between the physics-informed framing and actual neuroscience grounding, (2) the reliance on small sample sizes (particularly PPMI with 90 subjects) without cross-validation, (3) the lack of evidence that the circuit analogy is more biologically valid than alternative communication models, and (4) the risk of over-interpreting post-hoc interpretability analyses as causal neuroscientific mechanisms. A dedicated limitations section would significantly strengthen the paper.

**Strengths And Weaknesses:**

**Strengths:**

**S1. Novel physics-informed attention mechanism.** Formulating SC-FC fusion as an information flow routing problem and deriving a closed-form solution (Eq. 10) that unifies local edge capacity, global structural topology, and functional demands into a single differentiable expression is a creative and mathematically elegant design. Using circuit-theoretic inductive bias to reparameterize attention, enforcing global equilibrium and multi-path propagation, which is a meaningful architectural contribution that distinguishes this work from prior fusion strategies. That said, the true potential of this mechanism may be better demonstrated on larger-scale, general graph learning benchmarks where performance differences are more reliable and the confounding effects of small sample sizes and neuroimaging preprocessing variability are less pronounced.

**S2. Rigorous theoretical derivation.** The closed-form derivation of total information flow (Theorem B.1 in Appendix B) is mathematically sound and clearly presented. The connection from Kirchhoff's laws through the incidence matrix decomposition to the final quadratic form is well-motivated, and the computational complexity analysis (Appendix C) addressing the practical tractability of the Laplacian inverse is a useful addition.

**S3. Comprehensive baselines.** Comparing against 14 methods spanning general graph learning (MLP, GCN, GAT, GIN, GraphSAGE, Graphormer) and specialized brain network models (BrainGNN, Triplet, Cross-GNN, RH-BrainFS, NeuroPath, etc.) provides a thorough competitive landscape.

**Weaknesses and Concerns:**

**W1. The paper's core motivation is contradictory.** The paper criticizes prior methods for "lacking fundamental neuroscientific insight," yet AFR-Net itself is explicitly a physics-informed framework based on electrical circuit theory (Kirchhoff's laws, effective resistance, Ohm's law) as stated in its own title, which belongs to physics and engineering, not neuroscience. Actually, several of the paper's own baselines are arguably closer to being neuroscience-informed: NeuroPath models topological detours grounded in multi-pathway neural signaling, and RH-BrainFS explicitly addresses regional heterogeneity of SC-FC coupling, a well-documented neuroscientific phenomenon. The authors should seriously reconsider the core motivation. So if the insights come from neuroscience, experimental results should show how this method is embedded with a neuro-inspired prior or how the experiments align with some rigorous neuroscience findings. Otherwise, it risks being just another technical combination for multimodal graph learning.

**W2. Technical contribution lacks sufficient experimental rigor to stand on its own.** Even evaluated purely as a technical contribution for embedding a circuit-theoretic solver into a differentiable framework, the experimental validation falls short. The paper applies a single hyperparameter configuration without clarifying whether baselines were individually tuned, uses a fixed 6:1:3 split with only 3 seeds instead of cross-validation (particularly problematic for the small PPMI dataset), and omits key details on how baselines receive their SC/FC inputs. Classification results can vary significantly with different preprocessing techniques and across different subjects, which is a major difference between graph learning in neuroimaging and other graph data (social networks, recommendation systems, etc.). Applying it to brain network fusion is reasonable, but requires substantially more rigorous experiments to demonstrate genuine added value, such as 5-fold or 10-fold cross-validation, which the current evaluation does not provide.

**W3. The flow routing module is a new attention mechanism with additional learnable parameters without further supportive experiments.** Despite the physics framing, AFR-Net's core flow routing module ultimately produces edge-level importance scores $\Phi_{ij}$ that bias a transformer's attention. Basically, it is a new way to compute attention weights. The edge capacities $c_{ij}$ driving the entire computation are learned end-to-end via a neural network gate, not derived from any neuroscientific prior or physical measurement. The circuit formula thus serves as a differentiable reparameterization of attention, not as an injection of genuine domain knowledge. This raises a fundamental question the paper does not address: does the improvement come from the circuit-theoretic inductive bias, or simply from introducing additional learnable parameters (the edge capacity matrix $\mathbf{C}$)? Standard attention mechanisms in the baselines do not have per-edge learnable parameters of this kind. If one were to augment a vanilla transformer with the same number of learnable edge parameters using a simpler parameterization (e.g., a directly learned attention bias matrix), would the performance gap disappear?

**W4. The interpretability analysis is superficial and does not support the neuroscience claims.** The paper positions interpretability as a major contribution, claiming AFR-Net is "a mechanistic instrument for neuroscientific discovery." However, the analysis has fundamental issues. First, the structural core finding is trivial, since visual and somatomotor networks inherently have the densest white matter fiber bundles and highest SC weights, so any SC-weighted flow computation will naturally assign them the highest flow. Comparison to a null model (e.g., raw SC degree centrality) is necessary to support this claim. Otherwise, it is a consequence of input data statistics, not a model discovery. Second, post-hoc literature alignment is not validation, and the neuroscience literature on anxiety and OCD reports abnormalities across nearly all major brain networks, so virtually any pattern the model produces can be matched to some published finding. Comparison results for baseline models are also necessary. Third, the Neurosynth functional decoding is over-interpreted, cause the hippocampus–episodic memory association is textbook-level neuroscience. Using Neurosynth to rediscover this, then building a causal mechanistic narrative from what is purely statistical co-occurrence, represents a significant inferential leap. At least two different datasets for the same task should cross-validate each other. For an ICML-level paper making strong neuroscience claims, the interpretability analysis must go beyond post-hoc storytelling and requires quantitative, reasonable evidence with proper neuroscience-grounded controls, which the current evaluation entirely lacks.

**W5. The "issue" of regional heterogeneity is not convincingly shown to be unsolved.** The paper claims existing methods "fail to uncover the latent interactions between neural regions" and "cannot explain why SC and FC exhibit dynamic states of both coupling and heterogeneity." However, RH-BrainFS explicitly addresses regional heterogeneity, and NeuroPath models topological detours. The paper doesn't provide a clear empirical demonstration of where these methods fail mechanistically, while it only shows that AFR-Net gets higher accuracy and may also get lower AUC. A direct comparison showing, for instance, that the flow patterns learned by AFR-Net capture something the baselines' attention maps miss would strengthen this claim considerably.

**Minor Issues:**

- **The framing of "macroscopic cognitive phenotypes" is mismatched with the actual experiments.** The paper opens with the ambitious motivation of understanding how microscopic neuronal connectivity gives rise to macroscopic cognitive phenotypes. However, this term in neuroscience typically refers to a broad, systematic concept encompassing continuous cognitive dimensions such as intelligence, working memory capacity, executive function, and emotional regulation. The paper's actual downstream tasks are limited to binary disease classification (patient vs. healthy control), which is a considerably narrower scope than what "cognitive phenotypes" implies. To improve the motivation, the authors should at least include regression tasks on continuous cognitive scores.

- **Taking absolute values of FC lacks justification.** The model uses $|A_{fc}|$ as communication demand (Eq. 10), treating negative correlations (anti-correlations) identically to positive ones. In neuroscience, these reflect fundamentally different mechanisms, like anti-correlations between DMN and task-positive networks, for instance, represent inhibitory or anti-phase relationships, not equivalent information exchange. This design choice may be reasonable, but it requires explicit discussion and justification, which the paper does not provide.

---

> ### Author Rebuttal · Authors · 2026-03-30
>
> **W1**
>
> We acknowledge "lacking fundamental neuroscientific insight" was an overstatement. A more precise motivation is: *"While some methods incorporate neuroscientific observations, there remains a lack of principled formulations that bridge structural connectivity and functional dynamics."* We will revise accordingly. Circuit theory and diffusion models are the dominant neuroscience framework for communication dynamics [1]. The distinction is between topological methods (NeuroPath, RH-BrainFS) and communication dynamics methods (AFR-Net, Figure 1).
>
> **W2 and Q3**
>
> We conducted 5-fold stratified CV on PPMI across 3 seeds (15 runs):
>
> | Method | F1 (5-fold CV) | AUC (5-fold CV) |
> |--------|---------------|-----------------|
> | AFR-Net | 82.90 ± 3.54 | 89.80 ± 1.56 |
> | AFR-Net (original) | 83.70 ± 3.85 | 90.38 ± 1.64 |
> | NeuroPath (original) | 74.56 ± 0.76 | 71.88 ± 1.13 |
>
> Results are consistent with original report and remain superior to NeuroPath. For baselines: all general methods use dual-stream encoding (FC and SC each encoded separately, concatenated); multimodal methods follow their original designs. ABCD (6,381 subjects) is robust with 3 seeds.
>
> **W3 and Q2**
>
> We ablated using raw SC as edge capacities ($c_{ij}=\text{SC}_{ij}$, no edge gate):
>
> | Variant | OCD F1/AUC | ADHD F1/AUC | Anx F1/AUC |
> |---------|-----------|------------|-----------|
> | w/o Flow Routing | 60.53/68.21 | 60.12/59.23 | 49.70/58.00 |
> | Raw SC Capacity | 57.31/65.64 | 55.87/55.03 | 55.63/51.12 |
> | Full Model | **70.62/72.68** | **65.50/62.38** | **62.15/59.03** |
>
> The framework and adaptive learning are synergistic: the circuit formulation provides the inductive bias, while learned gating adapts capacities to functional context — both essential. Crucially, C is constrained by $L_{flow}=B^\top CB+\delta I$: changing one edge capacity affects *global* flow distribution, unlike standard per-edge attention weights which act independently. This global coupling fundamentally limits the hypothesis space.
>
> **W4**
>
> We respectfully disagree that flow patterns trivially follow SC weights. $Φ_{ij}$ (Eq. 10) is jointly determined by edge capacity, global topology ($L_{flow}^{-1}$), and functional demands ($L_{fc}$ from FC) — high SC with low FC demand yields low flow. If SC alone determined patterns, GCN on SC should perform well (it underperforms MLP, Table 1), and Anxiety/OCD should show identical patterns — instead, AFR-Net discovers distinct circuits (Hippocampus-DMN vs Visual-Limbic).
>
> We conducted null model comparison (top-100 edges, 3 checkpoints):
>
> | Null Model | Anx Jaccard | OCD Jaccard |
> |-----------|------------|------------|
> | SC edge weight | 0.023 | 0.022 |
> | SC degree product | 0.000 | 0.000 |
>
> Near-zero overlap (~2%) confirms non-trivial discovery. The group differences in Figure 3(b) (p<0.05, FDR) cannot arise from SC alone — SC does not differ between patients and controls at population level. Cross-disease validation further strengthens this: Anxiety (Hippocampus-DMN, Figure 3) and OCD (Visual-Limbic, Figure 5) reveal distinct circuits from the same structural substrate.
>
> **W5**
>
> We do not claim prior methods entirely fail — rather, they address heterogeneity at the topological level (e.g., RH-BrainFS uses fusion bottlenecks, NeuroPath uses detour masks). AFR-Net complements these by addressing heterogeneity at the communication dynamics level: the flow solver explicitly computes how much each edge contributes to global information transfer, naturally producing region-specific coupling patterns. Empirically, Table 1 shows that local aggregation methods (GCN, GAT, GIN) consistently underperform even MLP on brain networks, confirming the need for global modeling approaches.
>
> **Q1**
>
> The discrepancy reflects a recall-precision trade-off. AFR-Net achieves substantially higher recall (71.67% vs 69.17%) at the cost of slightly lower precision, resulting in higher F1 but lower AUC. In clinical neuroimaging, higher recall (fewer missed diagnoses) is generally preferred, as false negatives carry greater clinical cost than false positives. Moreover, the 1.8% AUC gap (62.38 vs 64.19) falls within the standard deviation range (±1.08 vs ±0.35), and ADHD is the most challenging task where all methods perform near chance level.
>
> **Q4**
>
>  |$A_{fc}$| models communication demand magnitude: both positive correlations and anti-correlations require information transmission through the structural substrate. This follows the communication dynamics literature [2]. We will add this discussion.
>
> *Reference*
>
> [1] Avena-Koenigsberger A, Misic B, Sporns O. Communication dynamics in complex brain networks. Nat Rev Neurosci. 2017 Dec 14;19(1):17-33. doi: 10.1038/nrn.2017.149. PMID: 29238085.
>
> [2] Seguin C, Tian Y, Zalesky A. Network communication models improve the behavioral and functional predictive utility of the human structural connectome.

---

### Decision · Program_Chairs · 2026-04-30

**Decision:**

Accept (regular)

**Comment:**

The paper introduces AFR-Net, a framework that couples structural connectivity (SC) and functional connectivity (FC) by treating SC-FC coupling as an information flow routing problem with a physics-informed or circuit-inspired formulation. The method is designed to plug into existing backbone networks for downstream prediction tasks on brain graphs, and the paper emphasizes interpretability of the routing patterns as latent communication structure.

Reviewers liked the mathematical clarity of the routing story and the fact that the module yields interpretable quantities that align with how clinicians and network neuroscientists talk about flow. Empirically, the paper reported gains when the routing module is added, and the rebuttal strengthened robustness claims with cross-validation on at least one key cohort and additional ablations (null models, fixed-capacity baselines, etc.). Several reviewers explicitly praised the baseline fairness discussion and the additional experiments that separate “learned routing” from trivial SC-weighting shortcuts.

Early concerns included: cubic scaling when naively applied at very fine granularities, need for clearer limitations on high-resolution use, and requests for stronger statistical validation across folds. The authors responded with CV results, additional analyses, and clarifications. Three reviewers posted fully positive acknowledgements and maintained their positive scores. One reviewer scored borderline rejection in the initial round and did not post a final acknowledgement in the export, so that concern should still be treated as a remaining caveat.

Reviewer feedback still asks for an explicit limitations section about region-level analysis versus voxel-level costs, and authors committed to add it.

Despite the remaining low initial score and the need for a crisp limitations paragraph, the overall evidence after rebuttal supports acceptance. Given these strengths, we accept.